# The F-pilus biomechanical adaptability accelerates conjugative dissemination of antimicrobial resistance and biofilm formation

Jonasz B. Patkowski [1,6], Tobias Dahlberg[2,6], Himani Amin [1], Dharmender K. Gahlot [3], Sukhithasri Vijayrajratnam[4], Joseph P. Vogel [4], Matthew S. Francis[3], Joseph L. Baker [5] ✉, Magnus Andersson [2] ✉ & Tiago R. D. Costa [1] ✉

Conjugation is used by bacteria to propagate antimicrobial resistance (AMR) in the environment. Central to this process are widespread conjugative F-pili that establish the connection between donor and recipient cells, thereby facilitating the spread of IncF plasmids among enteropathogenic bacteria. Here, we show that the F-pilus is highly flexible but robust at the same time, properties that increase its resistance to thermochemical and mechanical stresses. By a combination of biophysical and molecular dynamics methods, we establish that the presence of phosphatidylglycerol molecules in the F-pilus contributes to the structural stability of the polymer. Moreover, this structural stability is important for successful delivery of DNA during conjugation and facilitates rapid formation of biofilms in harsh environmental conditions. Thus, our work highlights the importance of F-pilus structural adaptations for the efficient spread of AMR genes in a bacterial population and for the formation of biofilms that protect against the action of antibiotics.

Conjugation is a sophisticated strategy employed by bacteria to exchange antimicrobial resistance (AMR) genes among bacterial populations. This process is one of the major contributors to the sharp increase in infections caused by pathogens harbouring plasmids that encode AMR genes[1–3]. At the molecular level, the transport of plasmid DNA in Gram-negative bacteria relies on a type IV secretion system (T4SS) to elaborate extracellular conjugative pili. These helical appendages are critical to establish and maintain direct contact between two mating cells before the transport of plasmid DNA is initiated[4–6]. One remarkable example of conjugative pili is the F-pilus, which is anchored to the T4SS via the outer-membrane core complex

(F-OMCC)[7]. Recently, the atomic structure of the F-OMCC was solved, and the highly dynamic properties of the complex and its capacity to accommodate cycles of extension and retraction of the F-pilus filament were revealed[8,9]. The available structures of the F-pilus from *Escherichia coli* and *Klebsiella pneumoniae* revealed that the filament is constructed by units of phosphatidylglycerol (PG) phospholipids and F-pilin interacting in a 1:1 ratio[10,11]. Strikingly, the dimensions of the F-pilus lumen are well suited to accommodate the passage of single-stranded DNA (ssDNA), and its mildly negatively charged lumen generates an ideal electrostatic environment that could aid the transport of ssDNA through its interior[10,12].

[1]MRC Centre for Molecular Bacteriology and Infection, Department of Life Sciences, Imperial College, London SW7 2AZ, UK. [2]Department of Physics, Umeå University, 901 87 Umeå, Sweden. [3]Department of Molecular Biology, Umeå University, 901 87 Umeå, Sweden. [4]Department of Molecular Microbiology, Washington University School of Medicine, St. Louis, MO 63110, USA. [5]Department of Chemistry, The College of New Jersey, Ewing, NJ 08628, USA. [6]These authors contributed equally: Jonasz B. Patkowski, Tobias Dahlberg. ✉e-mail: bakerj@tcnj.edu; magnus.andersson@umu.se; t.costa@imperial.ac.uk

Among different conjugative plasmids, IncF plasmids (encoding F-T4SSs) show high rates of conjugation in vitro and are the most frequent type of plasmid present in bacteria isolated from human and animal sources[13–15]. IncF plasmids are present in common enteropathogenic species such as *E. coli*, *Salmonella enterica*, *K. pneumoniae* and *Shigella* spp. and are able to mobilise within the mouse gut microbiome at rates similar to those observed in biofilms formed under in vitro conditions[16]. Additionally, the prevalence of IncF plasmids in bacterial isolates from urine[17–20] and blood samples[21–24] may suggest that IncF plasmids could benefit from adaptations that allow for dissemination despite high hydrodynamic forces present in those liquid environments. Moreover, F-pili also play an important role in biofilm formation[25]. In the early stages of biofilm formation, the F-pilus is directly responsible for cell-to-cell interactions, and the deletion of the gene encoding F-pilin dramatically decreases biofilms on the surface of epithelial cells, thereby impairing the infectious potential of adherent-invasive *E. coli* in vivo[26]. Therefore, we hypothesise that the unique structural properties of the F-pilus must grant high levels of physical stability to maintain contact between two cells to ensure both successful DNA transport during conjugation and the formation of a protective biofilm community through cell aggregation.

In this study, we characterised the underlying biomechanical properties of the F-pilus that ensure successful delivery of DNA during conjugation. We show that the same F-pilus properties also accelerate the bacterial biofilm formation capability. We demonstrate the extraordinary suitability of the conjugative F-system to operate in highly dynamic environments and show that the observed high stability of the mating pair between donor and recipient cells relies on the operation and structural adaptations of the F-pilus. We characterise the biomechanical properties of the F-pilus as a polymer highly resistant to bending and stretching, and reveal the role of the phospholipid molecules as an essential structural adaptation that provides exceptional robustness to the filament. Taken together, these properties of the F-pilus equip the bacteria with an unexpected dual strategy to equally potentiate the transmissibility of multidrug-resistant IncF plasmids and the formation of protective biofilm communities in the environment.

## Results

### F-pili enhance conjugative DNA transmission under hydrodynamic stress

It is unclear how the mechanical stress experienced by the bacteria in the environment or human gut affects the formation of mating bridges and thus bacterial conjugation efficiency. To test the mechanical properties of the F-pili in conditions closer to the natural bacterial environment, a set of conjugative assays were conducted in increasing mechanical stress conditions to mimic some of the conditions experienced by bacteria in the environment. For mating, we tested two strains of *E. coli*—the laboratory strain DH5α, and the enterohaemorrhagic *E. coli* (EHEC) clinical isolate O157:H7 strain EDL933. Donor strains (designated as F⁺) harbour the conjugative pOX38 plasmid that belongs to the IncF family of conjugative plasmids and is a reduced form of the canonical F plasmid, generated by HindIII digestion and circularisation of the largest fragment that retains the conjugative ability, additionally a chloramphenicol resistance cassette was also incorporated[27]. Donors were incubated with the recipient strain *E. coli* DH10β. As a control, the conjugation reaction was also conducted in the same way with plasmid-cured strains (designated as F⁻). Conjugation was conducted at 37 °C for 30 min in four separate groups with increasing hydrodynamic and mechanical stress: in steady media without shaking, in media with shaking at 180 rpm, in media with shaking at 180 rpm while supplemented with 50% (w/v) glass beads, and in steady media without shaking but with vortexing for 1 min at 4 min intervals (Fig. 1A). Conjugation efficiency was calculated as a

ratio of resulting transconjugants in the population over transconjugants plus viable recipients (T/T + R). The results showed a significant increase in the number of transconjugants when conjugation was performed in media with shaking when compared to steady media in both DH5α and EDL933 (Fig. 1B). Typical resulting plates in those experiments were also presented in Supplementary Fig. 1A, B. Interestingly, the additional hydrodynamic and mechanical stress provided by the presence of glass beads in the media during mating did not affect the conjugation efficiency when the resulting number of transconjugants was compared with mating without glass beads being present (Fig. 1B). Moreover, and highly surprisingly, when the mating pairs were subjected to cycles of extreme hydrodynamic stress defined by 1 min of vortexing followed by 4 min of steady incubation, the conjugation ability was not affected when compared with the steady incubation (Fig. 1B).

### F-pilus-mediated biofilm formation is enhanced in dynamic environments and donor-recipient cell populations

Cells harbouring conjugative plasmids can also produce biofilms; however, the role of the recipient cell and the presence of environmental flow forces in the development of the biofilm communities are unknown. As the F-pilus is responsible for cell-to-cell contact between bacteria, it is plausible that F-pili evolved to not only provide adhesion to ensure DNA delivery but also to function as a "sterile" adhesion appendage, i.e., they are incapable of conjugative transfer but assist in establishing donor-recipient biofilm communities. To understand how hydrodynamic stress affects the formation of biofilms, *E. coli* DH5α and EDL933 with (F⁺) and without (F⁻) the F plasmid pOX38 were incubated with 10 times excess of a recipient strain *E. coli* DH10β. The incubation proceeded under three different conditions—in a steady environment without shaking, in an environment with shaking and in a shaking environment with glass beads—to mimic increasingly dynamic conditions. The biofilms were allowed to develop for 24 hours at 37 °C. Following incubation, planktonic cells were decanted from the glass tubes, leaving visible biofilms on the air-liquid interface in the F⁺ populations with shaking and glass bead cultures and very weak biofilms in all F⁻ cultures. Quantification of biofilm production was assessed with crystal violet staining and subsequent absorbance measurements at 595 nm (Fig. 1C). These results corroborated the conjugation efficiency assays, where higher hydrodynamic stress enhanced conjugation efficiency (Fig. 1B, C). The plasmid-cured DH5α(F⁻) and EDL933 (F⁻) cells formed weak biofilms (Fig. 1C). In contrast, DH5α(F⁺) and EDL933 (F⁺) developed pronounced biofilms (Fig. 1C). Like in our conjugation experiments, formation of biofilms was significantly enhanced by an increase in mechanical stress. Furthermore, whole-bacteria negative-stain transmission electron microscopy (TEM) images show an extensive network of F-pili connecting bacterial cells, which likely contributes to the formation of the biofilm matrix (Fig. 1C, inset). Our hypothesis is that higher conjugation efficiency and biofilm formation observed in bacteria exposed to higher hydrodynamic stress conditions are due to a higher probability of encountering between donor and recipient cells in the environment. Therefore, we used an f1 bacteriophage infection to test this hypothesis. The f1 bacteriophage is a filamentous phage that attaches specifically to the F-pilus tip. Any potential alteration in the f1 phage infection rate under varying mechanical stress would indicate effects on the successful attachment of the phage to the F-pilus, analogous to that of a recipient bacterium to the F-pilus, as both processes are subject to changes in chance of encounter due to mixing. One DH5α(F⁺) cell culture was infected with f1 phage, split into 2 vials and incubated for 30 min at 37 °C—one vial was incubated in steady media and the other with shaking at 180 rpm. There was an obvious increase in plaque formation in the sample incubated under shaking conditions (Supplementary Fig. 1C). As anticipated, elevated flow forces accelerated phage attachment to the F-pili, resembling

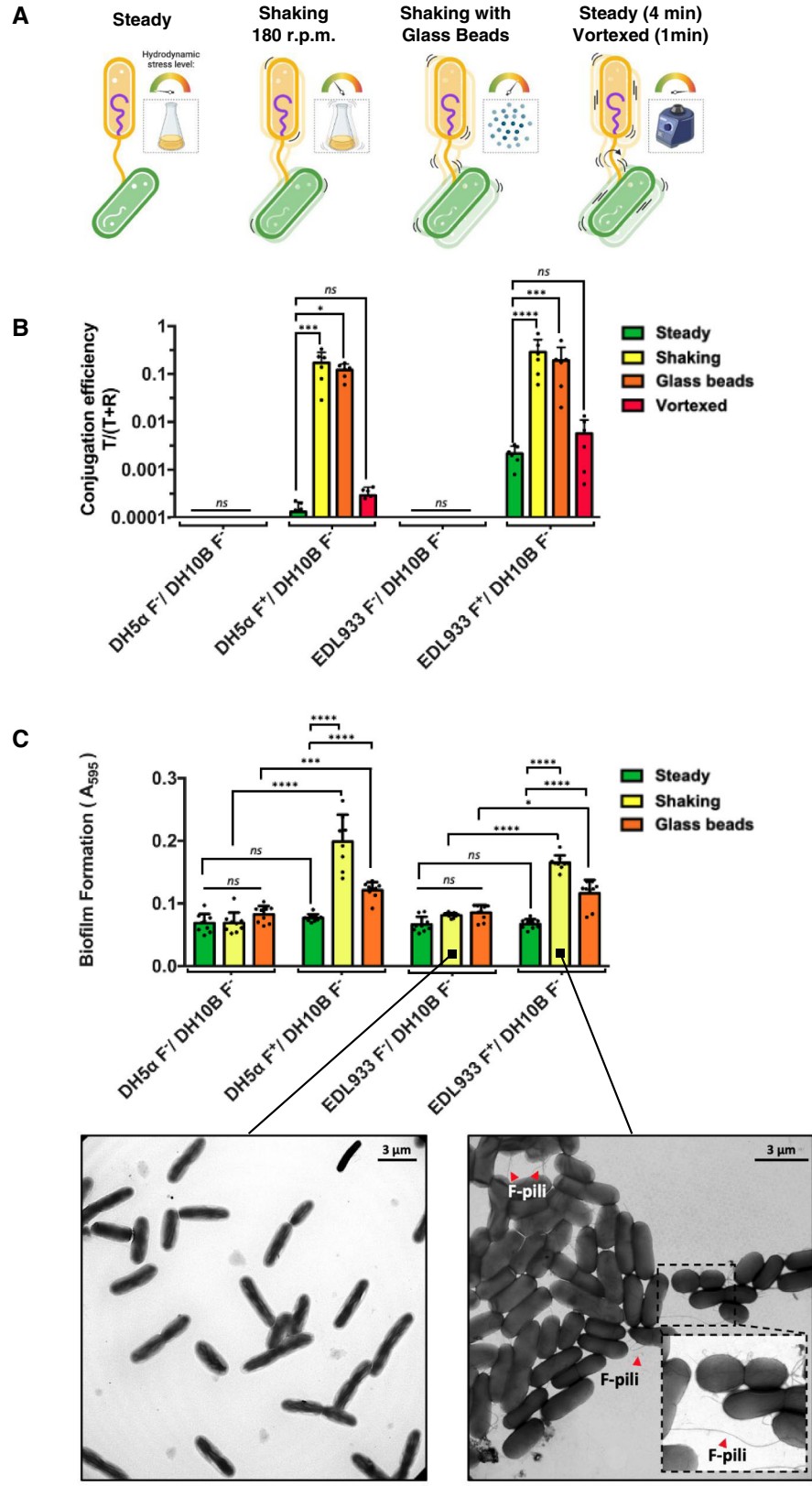

the results obtained for conjugation and biofilm formation in corresponding hydrodynamic conditions. These results, together with our TEM images of bacteria within the biofilm matrix, suggest that it is the operation of the F-pilus that accounts for the higher conjugation efficiency and biofilm formation ability observed in dynamic media.

## F-pili are flexible and thermochemically stable appendages

We show that conjugation and biofilm formation are more successful in dynamic environments. Therefore, the F-pilus structure potentially evolved to retain unique biomechanical properties that enable filament operation, i.e., to promote conjugation and create biofilms in dynamic environments. Additionally, extremophilic bacteria are also

**Fig. 1 | Effect of mechanical stress on conjugation efficiency and biofilm formation. A** Graphical representation of the tested conjugation conditions. From the left: conjugation conducted in steady media (without shaking); conjugation conducted in media shaking at 180 rpm rate; conjugation conducted in a shaking media with glass beads of 2–3 mm in diameter; steady media (without shaking) interrupted by vigorous vortexing for 1 min every 4 min. **B** Ratio of transconjugants (T) over transconjugants + recipients (T + R) obtained after 30 min of incubation of two donor strains DH5α (F⁺) and EDL 933 (F⁺) and two plasmid-cured strains DH5α (F⁻) and EDL 933 (F⁻) with recipient strain DH10β(F⁻) in all four conditions described above—steady (green), shaking at 180 rpm (yellow), supplemented with Glass Beads (orange) and vortexed (red). All *p* values were derived from comparisons of each condition via two-way ANOVA with Tukey's correction for multiple comparisons. DH5α F⁺/DH10B F⁻ steady vs shaking ***$p$ = 0.0008 and steady vs glass beads *$p$ = 0.0243; EDL933 F⁺/DH10B F⁻ steady vs shaking ****$p$ = 0.0001, steady vs glass

beads ***$p$ = 0.0002. Data were obtained from n = 6 biological replicates. **C** Biofilm formation ability obtained after 24 h growth of two donor strains DH5α (F⁺) and EDL 933 (F⁺) and two plasmid-cured strains DH5α (F⁻) and EDL 933 (F⁻) with recipient strain DH10β(F⁻), as described for conjugation ability. All p-values were derived from comparisons of each condition via two-way ANOVA with Tukey's correction for multiple comparisons. DH5α F⁻/DH10B F⁻ glass beads vs DH5α F⁺/DH10B F⁻ glass beads $p$ = 0.0003; EDL933 F⁻/DH10B F⁻ glass beads vs EDL933 F⁺/DH10B F⁻ glass beads *$p$ = 0.0106. All $p$ values denoted as **** were $p$ < 0.0001. Data were obtained from $n$ = 3 biological replicates measured in triplicate. Negative stain images for chosen conditions were presented below as an example of the F-pilus mediated cell aggregation. ****$p$ < 0.0001; ***$p$ < 0.001; **$p$ < 0.01; *$p$ < 0.05; ns not significant. Bars in all graphs represent mean values, whiskers represent standard deviation, and black circles represent each replicate value. **A** cartoons were created with BioRender.com. Source data are provided as a Source data file.

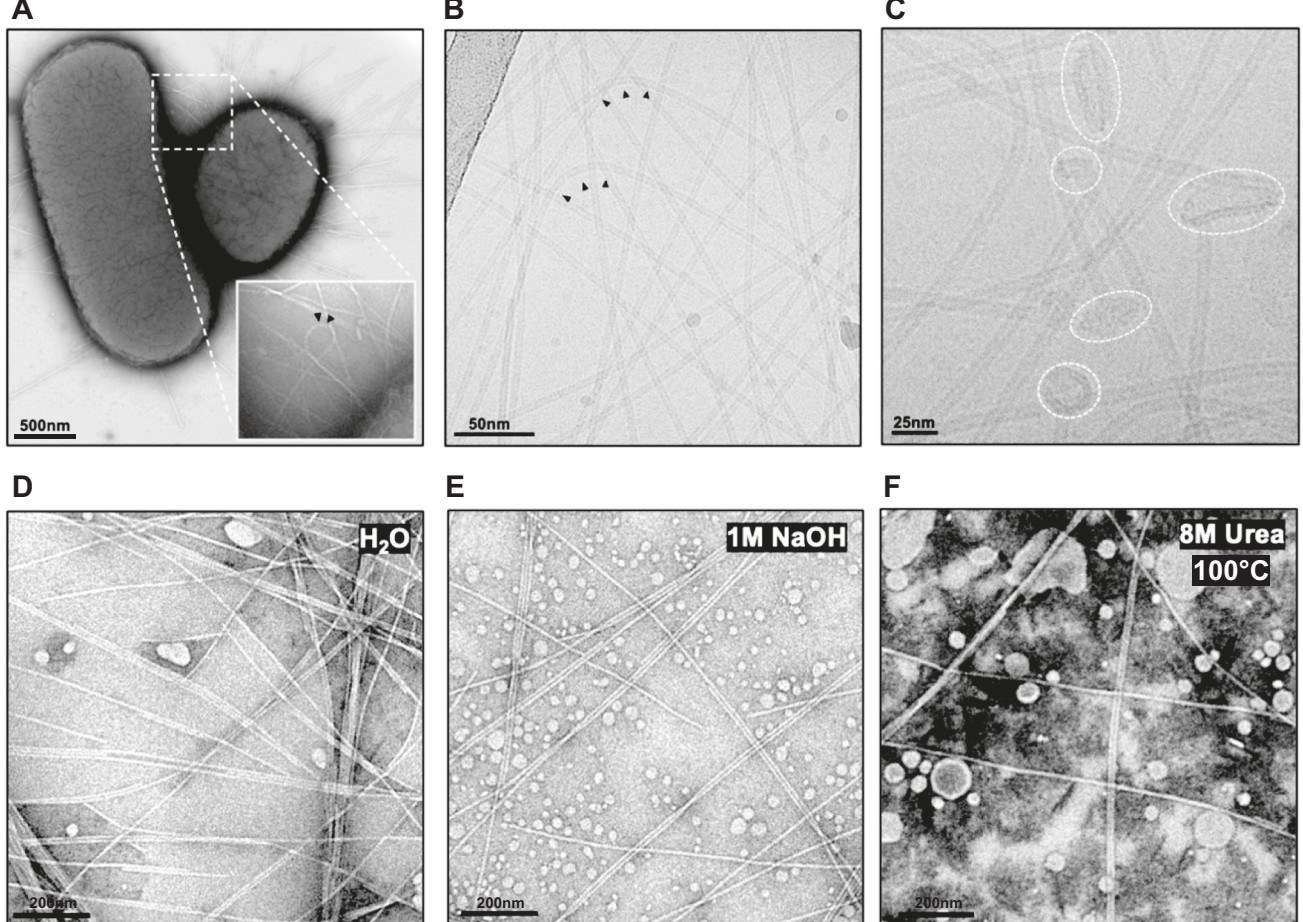

**Fig. 2 | Electron microscopy micrographs of intact bacteria and purified F-pili.**
**A** Negative stain EM micrograph of *E. coli* harbouring the pED208 F-plasmid.
**B** Cryo-EM micrograph of purified pED208 F-pili. **C** Cryo-EM micrograph showing portions of bacterial membrane at the cell proximal ends of the F-pili (white dashed circles). Arrowheads show bended pili at the bacteria surface and after cryo-EM

vitrification. F-pili exhibit significant structural resilience to harsh chemical and temperature treatments. **D** F-pili in water. **E** F-pili treated with 1 M NaOH. **F** F-pili treated with 8 M Urea and boiled for 30 min. Scale bars represent 200 nm. Each experiment was repeated separately a minimum of three times.

capable of conjugation under extreme environmental conditions, suggesting potential structural adaptations that provide thermo-chemical resilience.

To understand the shape of F-pili attached to bacterial cells, we first conducted negative-stain whole cell TEM imaging of intact *E. coli* DH5α(F⁺) harbouring the pED208 plasmid. This plasmid belongs to the IncF family isolated from *Salmonella typhimurium*[28] and expresses *tra* genes constitutively. For that reason, it was suitable for whole cell imaging, F-pili purification, and force measurements, but not for our

conjugation and biofilm formation analysis. The whole cell imaging revealed numerous F-pili filaments at the bacterial surface (Fig. 2A). Most of the filaments project straight out from the bacterial surface, but some displayed striking curvature to nearly 180 degrees without any signs of breakage (Fig. 2A, arrows inset). To examine this property in greater detail, F-pili were sheared off from the bacterial cell surface and purified. F-pili purity was confirmed by SDS–PAGE (Supplementary Fig. 2A), and the presence of TraA F-pilin was further verified by LC–MS/MS. Upon purification, the integrity and homogeneity of the F-pili

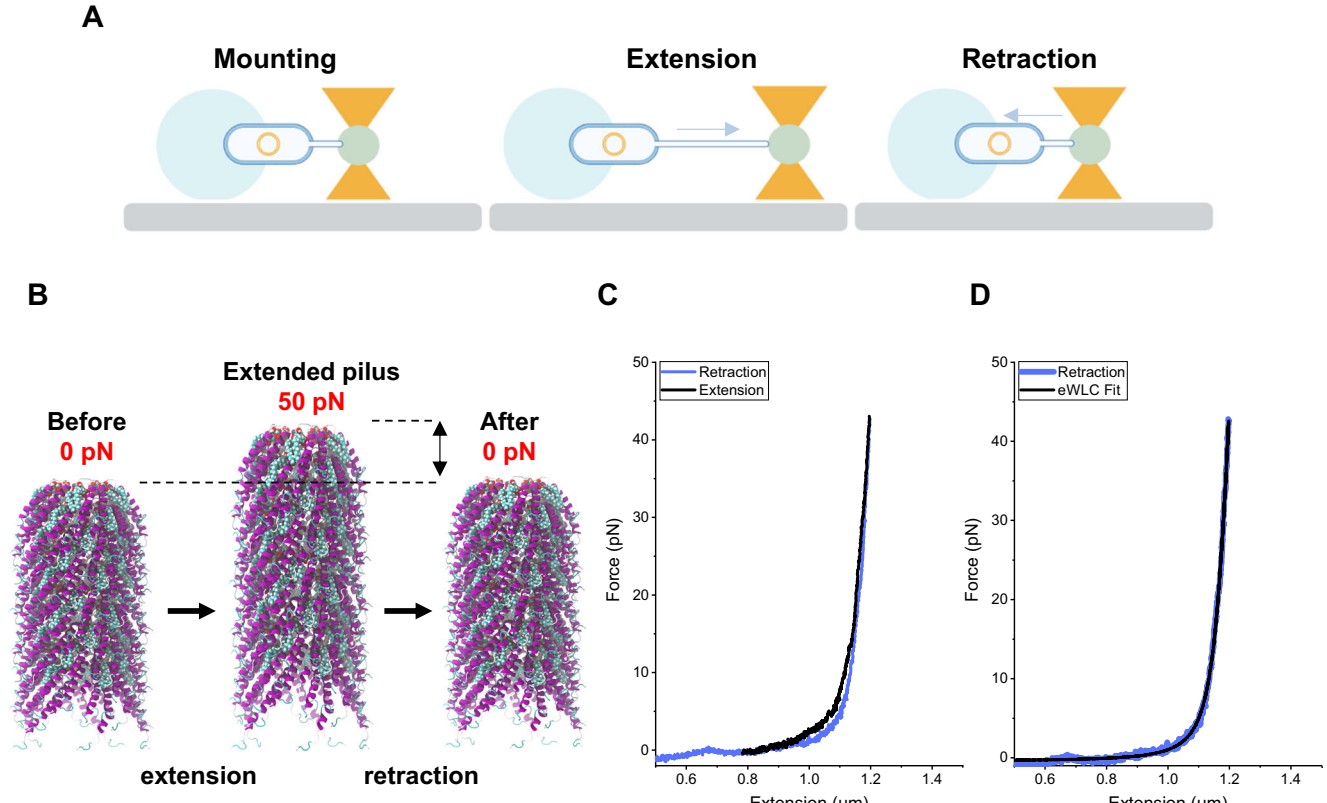

**Fig. 3 | The force-extension and contraction response of F-pili fits the eWLC model and shows no hysteresis. A** Overview of the experimental setup. A bacterial cell was mounted on the stage and the F-pilus was forcefully extended and retracted to measure force response. **B** Graphical representation of the result. Before the experiment starts, the pilus is in a relaxed state. After applying a force of 45 pN, the F-pilus extends under a non-linear force response. When the applied force is reduced, the pilus retracts back to its relaxed state. **C** Force extension (black) and retraction (blue) curves are overlapping with marginal hysteresis. **D** A force-extension curve (blue) fitted with the eWLC model (black). **A** cartoons were created with BioRender.com. Source data are provided as a Source data file.

were assessed by negative-stain EM before vitrification and cryo-EM imaging (Supplementary Fig. 2B). The cryo-EM micrographs from vitrified F-pili confirmed the ability of filaments to withstand significant bending without breakage (Fig. 2B, arrowheads). Further underlining the great mechanical resistance of the appendage was the presence of bacterial membranes at the base of the purified F-pili, suggesting that the appendage was more likely to carve parts of the membrane out of the bacterial cell envelope and then carry them along throughout the purification steps, rather than breaking apart during membrane extraction (Fig. 2C).

To investigate the thermochemical resistance of the F-pilus, purified fractions were subjected to extreme chemical and thermic conditions, followed by negative-stain EM to assess any resulting changes in morphology (Fig. 2D–F). F-pili displayed significant resistance to treatments with 1 M NaOH (Fig. 2E) or boiling in 8 M urea (Fig. 2F). These harsh conditions were not sufficient to disrupt filament integrity since their appearance still resembled pili recovered from untreated fractions (Fig. 2D). In all tested conditions, the F-pili preserved an intact and elongated morphology without any observable increase in the frequency of filament structure breakage. Only when the F-pili were subjected to the denaturing conditions of SDS–PAGE was it possible to visualise individual F-pilin subunits caused by disassembled appendages (Supplementary Fig. 2A). SDS molecules likely disrupt the protein-PG phospholipid interactions present in the F-pilus, which seemingly release the F-pilin subunits from the helical filament. The extraordinary robustness of the F-pili filament hints that it has evolved to operate despite vastly unfavourable conditions of the environment, including those with high mechanical and thermochemical stress. Since conjugation and

biofilm formation are central to bacterial survival and the biosynthesis of such large and complex appendages is likely to be a metabolically costly process, we show that the bacteria have developed a very durable polymer that ensures stable connections between interacting cells.

## Force-extension measurements using optical tweezers indicate an elastic filament

To assess the dynamics and elasticity of F-pili, we measured the biomechanical properties of F-pili using force measuring optical tweezers assays. These assays have proven efficient in measuring the biomechanical properties of other tube-like pili, such as Type-1, Pap, and CFA/I pili[29–31]. Using an optical tweezers instrument (Supplementary Fig. 4), we applied a tensile force to an F-pilus and assessed the force response with sub-pN resolution. An example of such an experiment is shown in Fig. 3A, and the obtained results are represented graphically in Fig. 3B. Without an applied force the pilus is in a relaxed state, but after a tensile force is applied the quaternary structure extends. The relative extension for a given force depends on the initial length of a pilus. When the external force is removed, the pilus retracts back to its relaxed state. A force-extension (black) and retraction (blue) response of an F-pilus are shown in Fig. 3C. There is only a marginal hysteresis in the force-extension curves since the retraction curve matches the extension curve well, indicating that the structure is intact and that the experiment is performed close to steady state. Even when performing multiple measurement cycles on a single pilus, no fatigue was seen despite the experiment lasting for several minutes. This result indicates that incubating the bacteria on ice stopped the polymerisation or depolymerisation of the pilus at the bacterial surface. Furthermore, we

could not observe any inelastic conformational changes in the pili for forces up to 100 pN. Force-induced conformational changes in the pilus would appear as either a sigmoidal (shifting spring constants of the pilus) or constant force versus extension response[32]. Thus, we conclude that F-pili have a pure elastic response for forces up to 100 pN, indicating no damage to the filament at the experimental timescale.

To describe the physical behaviour of F-pili and compare the properties to other pili types, we fitted our force data with an elastic worm-like-chain model (eWLC) model, Eq. (1). This model is suitable for describing semiflexible isotropic polymers. In this model, the polymer's entropic bending is given by the persistence length, and the enthalpic stretching is characterised by a stretch modulus. The retraction phase (blue) with a corresponding fit of the eWLC (black dashed curve) is shown in Fig. 3D. Thus, the fit provides the parameter values for the persistence length and stretch modulus, which, for these data, are 21 nm and 1357 pN, respectively. The average values of pili measured from 20 bacterial cells are $104 \pm 36$ nm (mean ± SE, median value 44 nm) and $2330 \pm 469$ pN (mean ± SE, median 1710 pN). We provide a full compilation of the estimated measured parameters in Table 1. The variation in these estimated model parameters, with outliers in the data set (Supplementary Fig. 5), can be explained by the possibility of two or more pili attaching to the probe bead or that pili bundle with each other. Bundling of pili has also been observed in other force studies with pili that exhibit short persistence length[33]. Despite this, the results suggest that F-pili are relatively bendable polymers. Furthermore, the stretch modulus reveals that F-pili are elastic and stretch in the axial direction by tensile force. This elasticity is seen in Fig. 3, where we can document that the force increases linearly after 1.2 μm extension up to approximately 45 pN, in which the extension is stopped. Since we did not have a covalent attachment of pili to the measuring bead, we could not apply significantly high forces, that is, hundreds of pN. However, 50-100 pN should be close to the maximum force that the F-pilus nanomachinery responsible for polymerisation is able to produce, as observed for Type IV pili[34]. Finally, we note that the persistence length of F pili is longer than for Type IV pili, which due to uncertainty, is reported as a range from 1 to 50 nm[33], whereas the spring constant is slightly lower than Type IV pili reported to 2.0 pN/nm[35].

**Molecular dynamics simulations reveal that phospholipids confer large structural stability to the F-pilus**

The structure of the F-pilus shows that the polymer is made of an equimolar number of F-pilin subunits and PG phospholipid molecules. Experimentally, it is not possible to evaluate the F-pilus structure in the absence of PG phospholipids, as the molecules are essential for cell viability – knocking out PGs entirely can be lethal to the bacterium[36]. Therefore, our simulations provide a unique and novel approach to obtain exclusive insight into the role that PG phospholipids play in the structural integrity of the F-pilus appendage. Hence, to evaluate the mysterious role of phospholipids in the F-pilus structure, we carried out both steered molecular dynamics (sMD) simulations and equilibrium simulations of the F-pilus 11mer (i.e., 11 layers of the F-pilus repeating pentamer) both with and without phospholipid molecules. In the sMD simulations, we pulled on the tip protein pentamer while restraining the pentamer at the base of the filament (see *Methods* for details). Movies of the sMD simulations for all five runs for each system (with and without phospholipids) are included in Supplementary Movies 1 and 2. As observed in the simulation trajectory movies, when phospholipids are present, the F-pilus is elongated but never breaks apart aside from individual proteins from the tip, eventually being extracted from the filament (Supplementary Movie 1). However, when phospholipids are removed from the structure, the F-pilus undergoes severe structural failure under force (Supplementary Movie 2). This

result suggests that phospholipids play an important role in the structural stability of the F-pilus. To quantify these differences, Fig. 4 shows the applied force versus simulation time for the simulations with and without phospholipids (Fig. 4A, B, respectively). For simulations including the PG phospholipids, the peak force corresponds to the extraction of tip proteins in the simulation runs (Fig. 4A and Supplementary Movie 1). However, when PG phospholipids are not included in the simulated model, a distinct peak feature is not observed (Fig. 4B), which is consistent with the structural failure observed in the simulation trajectories for the F-pilus without phospholipids (Supplementary Movie 2). Structural failure without phospholipids occurs at a significantly lower amount of total applied force compared to the peak force required to elongate the pilus and extract even a single protein from the F-pilus tip when phospholipids are present (Fig. 4A, B). Static snapshots at various time points throughout one of the sMD trajectories with and without phospholipids are also shown (Fig. 4C, D, respectively). We also observed significant differences in the stability of the F-pilus with and without phospholipids under equilibrium conditions, as evidenced by calculations of the root mean square deviation (RMSD) of alpha carbon coordinates (Fig. 5A, B). The RMSD calculations are taken with respect to the initial F-pilus structure (Fig. 5C). Whether we calculate the RMSD using all alpha carbon coordinates or if we exclude the alpha carbons from the tip and base pentamers, the observed trend remains the same. The structure of the F-pilus without phospholipids diverges significantly from the initial structure, while the F-pilus with phospholipids permits only a modest increase in the RMSD value over 100 ns equilibrium simulations. Furthermore, we have measured the secondary structure content of the initial F-pilus structure, as well as for the equilibrium simulations and the sMD trajectories (Supplementary Table 1). As expected, we observe that the helical content of the F-pilus remains high in the equilibrium simulation with lipids, and there is not a significant change in the helical content when lipids are removed. Also, the amount of helical content during the sMD simulations near the plateau of force (Fig. 5A, B) is also well-maintained (Supplementary Table 1), suggesting that the secondary structure of pilin subunits is resistant to external force on the filament regardless of the presence of the phospholipids. While secondary structure is well maintained, some individual interactions can become disrupted based on the presence or absence of phospholipids. In particular, we were interested in two salt bridges that occur between adjacent subunits, Glu29-Lys41, and Asp18-Lys64. The Glu29-Lys41 salt bridge is interior to the filament, while the Asp18-Lys64 salt bridges are at the exterior surface. Here we observe that without lipids presence, there is a sizeable decrease in the percentage time that the Glu29-Lys41 salt bridge persists over the equilibrium simulation with lipids, compared to without lipids (Supplementary Table 2). However, for the Asp18-Lys64 salt bridge, there is not a sizeable difference in the percentage time that the salt bridge is formed comparing the two equilibrium simulations, and the percentage is overall low in both cases (Supplementary Table 2). We also show a final snapshot from the end of the 100 ns equilibrium simulations of the F-pilus with phospholipids (centre) and without phospholipids (right), which demonstrate that phospholipids play an important role in the integrity of the F-pilus quaternary structure (compared to the initial structure in Fig. 5C). The equilibrium simulations for both systems demonstrate their dynamics over the 100 ns timescale (Supplementary Movie 3).

In summary, we demonstrated that phospholipid molecules are critical for stabilising pilin-pilin interactions that otherwise will be unable to maintain filament integrity on their own. Based on the drastic decrease in the robustness of polymers without phospholipids, we further hypothesise that incorporating phospholipids into helical filaments is likely a requirement to provide critical structural strength to other conjugative pili.

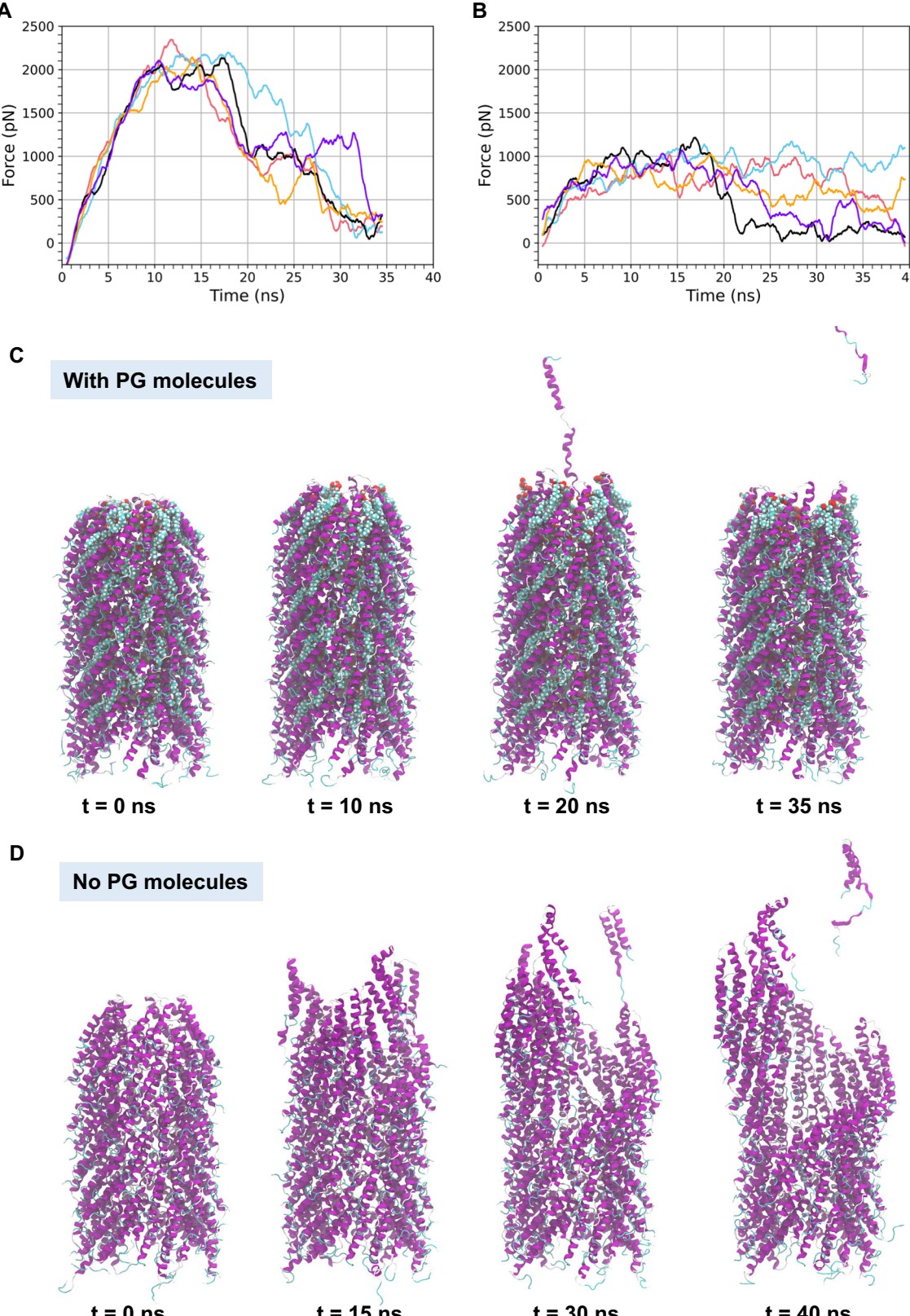

**Fig. 4 | The F-pilus force response depends on the presence of phospholipids.** Force versus time curves are shown for **A** the F-pilus simulations with phospholipids for each of the five runs (run 1−red, run 2−blue, run 3−black, run 4−orange, and run 5−purple) and **B** the F-pilus simulations without phospholipids. Four representative snapshots from a steered MD simulation (run 1) are shown for the F-pilus structure from the system **C** with phospholipids and **D** without phospholipids. The method to measure the force required to extend the F-pilus by pulling on the pentamer at the pilus tip is the steered MD approach, which is applied to the system as described in the "Methods" section. The force data plotted is a 1 ns running average. Source data are provided as a Source data file.

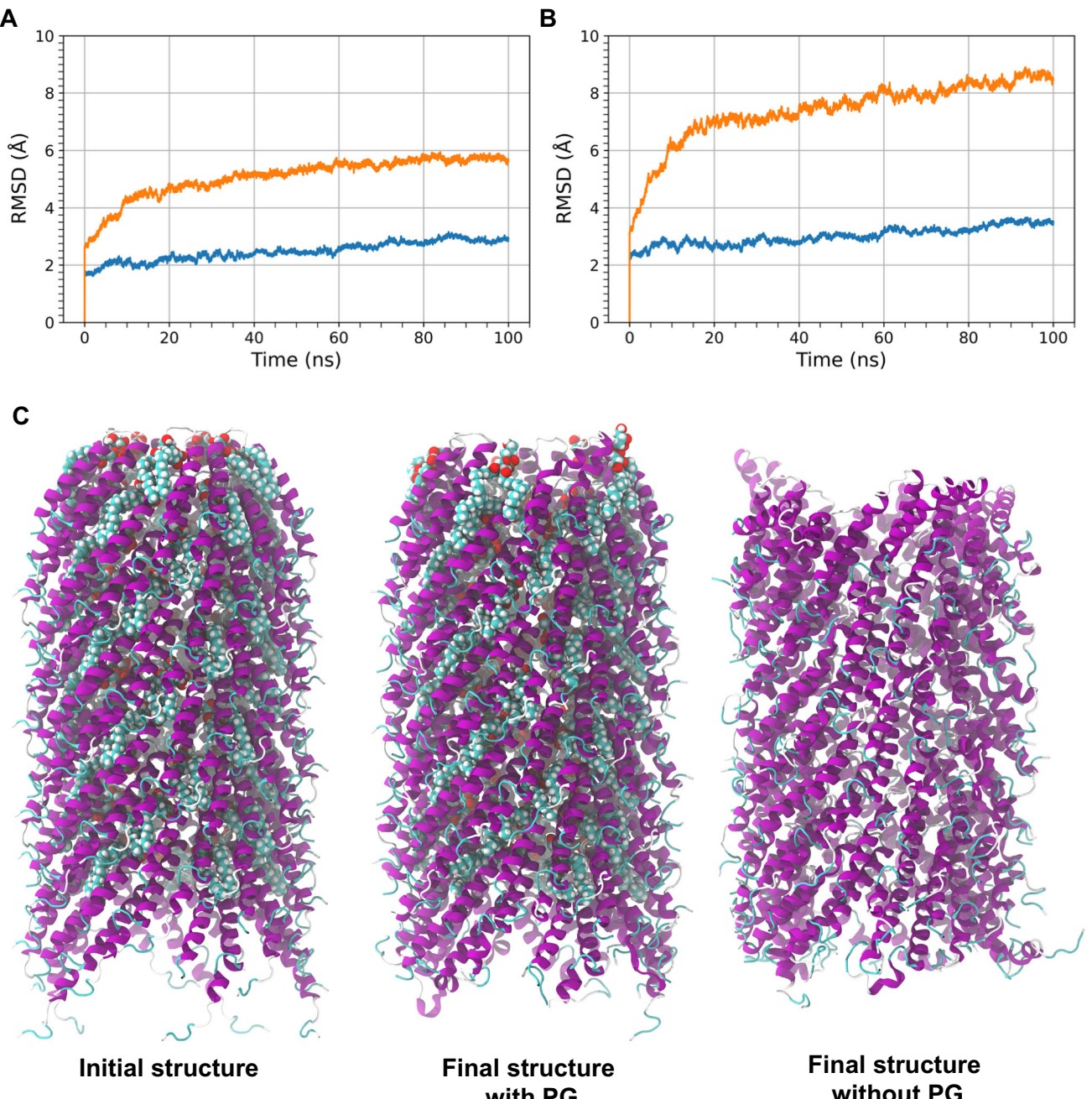

**Fig. 5 | Phospholipids provide structural stability to the F-pilus filament.** The root mean square deviation of the alpha carbon coordinates (calculated with respect to the initial filament structure) are shown for **A** alpha carbons excluding those belonging to the five subunits at the base and the five subunits at the tip of the filament and **B** all alpha carbons in the system. The RMSD calculations are performed on the 100 ns equilibrium simulations (with phospholipids−blue, without phospholipids−orange). **C** The initial F-pilus structure including phospholipids is shown at the left. The centre image shows the structure of the F-pilus with phospholipids from the end of the 100 ns equilibrium simulation. The rightmost image shows the structure of the F-pilus without phospholipids from the end of the 100 ns equilibrium simulation. Source data are provided as a Source data file.

## Discussion

The most widespread family of plasmids in human and animal isolates are those based on the IncF replicon[14]. These plasmids encode highly efficient conjugative machineries that evolved to ensure successful DNA transport in mechanically and chemically challenging environments, such as those encountered by bacteria in the human and animal gastrointestinal tract. One remarkable example of adaptation to complex environments is the intricate structural arrangement displayed by the multifaceted F-T4SS machinery. Made up of approximately 25 different proteins, the assembled machinery is capable of conjugating in a variety of extreme environments[37,38]. In

this work, we examined the impact of the structural adaptations of the F-pilus and their implications in maintaining a stable mating pair between donor and recipient bacteria to ensure efficient propagation of drug-resistance plasmids and formation of biofilm communities.

To mimic the conditions encountered by bacteria living in a host or free in nature, we exposed two *E. coli* strains harbouring an IncF plasmid; the common laboratory strain DH5α and a clinical isolate−the enterohaemorrhagic *E. coli* (EHEC) O157:H7 strain EDL933, which colonises the human colon and is responsible for large and severe outbreaks worldwide[39,40]−to a range of hydrodynamically and

mechanically challenging environments and probed their ability to conjugate and form biofilm communities.

We observed that bacterial connections established by the F-pilus are significantly resilient, which is in line with the observation that pOX38-mediated conjugation occurs at similar rates in broth and on agar surface[41]. Contrary to what was generally assumed, we observed that dynamic environments improve the search for mating opportunities. This could be due to the creation of a more even distribution between the donor and recipient bacterial populations, suggesting that bacteria utilise their well-adapted F-pili to exploit environmental flow forces and scan larger volumes of their surroundings while seeking an interaction partner. Furthermore, we were unable to forcefully disrupt the mating connections and hence abolish conjugation by applying extreme mixing forces. This observation implies that once the F-pilus from the donor cell contacts the surface of the recipient cell, the appendage establishes a highly stable connection that locks donor and recipient cells together. This strong interbacterial attachment mediated by the F-pilus surely provides an ecological advantage to the donor that goes beyond gene transfer and might explain the correlation observed between our conjugation and F-pilus-mediated biofilm formation assay results.

As the majority of bacterial infections are associated with biofilms and the cells present in these communities are up to 1000 times more resistant to antibiotics than their planktonic counterparts, this dual strategy of the F-pilus presents an obvious ecological benefit to the bacteria[42]. One could envisage that donor bacteria located at the boundaries of a three-dimensional biofilm could recruit novel members via their conjugative F-pili to reinforce biofilm structural integrity and accelerate its growth. This mechanism could also utilise the same dynamic properties of the F-pilus to exploit turbulent environment for more efficient scanning for an interaction partner. The new outer layers of biofilm formed by bacteria equipped with a newly acquired conjugative machinery, could be passed down from the inner layers of already established donor communities. This strategy would ensure that each new layer of biofilm cells maintains the same ability to utilise the very efficient F-pilus-mediated adhesion and can also further propagate the machinery needed to elaborate the F-pili. This is in line with the observation that only a fraction of F-pili is associated with a fully assembled conjugative T4S machinery in the membrane[7]. F-pili involved in nonconjugative interactions might be simply deposited in the membrane solely to maintain the established cell-to-cell connections. In this case, it is not necessary for them to remain associated with the secretion machinery that is needed only during conjugative contacts. Therefore, we propose that bacteria can assemble sterile F-pili incapable of mediating conjugation while maintaining their capacity to establish connections and develop a biofilm community.

In this study, we unveil the unique biomechanical properties of the F-pilus that explain its extraordinary suitability to create strong junctions between interacting cells in highly dynamic environments. We show that the F-pilus has the capacity to repeatedly accommodate mechanical stress without signs of fatigue and possesses significant structural resistance to bending, pulling, and stretching. The F-pilus also displays extraordinary resilience to harsh thermochemical conditions, which can be important for extremophilic bacteria to successfully mobilise IncF plasmids in their harsh niche[43].

Furthermore, we were able to attribute these unique biophysical and thermochemical properties to a single structural element - the presence of phospholipids in the F-pilus structure. Although it is not possible to evaluate the contribution of PG phospholipids in vivo[36], our sMD simulations clearly show their fundamental role in maintaining the integrity of the filament, including an effect on the Glu29-Lys41 salt bridge interaction, and how the filament more readily falls apart in the absence of the PG phospholipids under force, although secondary structure of individual pilin subunits is well-maintained regardless of the presence of absence of the PG phospholipids. Given the large

sequence and structural conservation among F-pilin (TraA) encoded by IncF plasmids (Supplementary Figure 3A and 3B), their extensive interaction interface[10,11], and the positive effect of phospholipid molecules on pilus structural stability, we postulate that other pili classes will also benefit from being structurally reinforced by the presence of phospholipids, as exemplified by the recent structures of minimised and archaeal conjugative pili[44,45].

Taken together, the exceptional architecture and biomechanical properties of the F-pilus ensure a robust connection between bacteria during conjugation and biofilm formation in virtually any environment. This phenomenon poses an obvious ecological advantage by guaranteeing rapid dissemination of drug-resistance plasmids in a wide range of environmental conditions while simultaneously accelerating propagation of protective biofilm communities in human and animal niches.

## Methods

### Conjugation efficiency assay

Both strains of *E. coli* harbouring the pOX38 plasmid (donor) – DH5α and EDL933 – were resuscitated from the frozen storage by streaking on a LB agar plate supplemented with 30 μg/ml chloramphenicol. *E. coli* DH10β strain (recipient) was streaked on a LB agar plate supplemented with 100 μg/ml streptomycin. Two separate cultures were started, supplemented with 30 μg/ml chloramphenicol and 100 μg/ml streptomycin for *E. coli* with pOX38 and *E. coli* DH10β, respectively, and allowed to grow overnight at 37 °C with shaking. On the next day, 500 μl of each overnight culture was used to inoculate two independent 25 ml cultures with appropriate antibiotics and growth was monitored until $OD_{600}$ reached 0.6 in both cultures. After, the cells were washed twice in 25 ml of chilled LB, the donor cells pellet was resuspended in 10 ml of chilled LB whereas the recipient cells were resuspended in 1 ml of chilled LB. In a separate pre-chilled tube, 1 ml of each resuspension was combined and thoroughly mixed by pipetting, generating a suspension with 1:10 donor to recipient ratio. 500 μl of the conjugation suspension was used to inoculate 4 different tubes filled with 4.5 ml of LB. Each tube was placed in the respective tested condition: (1) steady–tube was standing still without shaking; (2) shaking at 180 rpm; (3) shaking at 180 rpm and supplemented with 2.5 g of sterile glass beads (2-3 mm in diameter); (4) shaking at 180 rpm but interrupted with one-minute high-speed vortexing every four minutes. All conjugation reactions were allowed to proceed for 30 minutes at 37 °C. Conjugation reaction of plasmid-cured DH5α and EDL933 (F⁻) was conducted in a similar way, except for the addition of chloramphenicol to resuscitate the cells. To assess the number of transconjugants obtained in each condition, serial dilutions of each of the resulting conjugation reactions were prepared, ranging from $10^{-1}$ to $10^{-6}$, and 7 μl of each dilution was plated on LB agar supplemented with 30 μg/ml chloramphenicol and 100 μg/ml streptomycin. To count the total viable transconjugants and recipients, 7 μl of each dilution was also plated on LB agar supplemented with 100 μg/ml streptomycin. Colonies were counted for each condition and statistically compared using two-way ANOVA with Tukey's adjustment for multiple comparisons; performed and visualised using GraphPad Software (www.graphpad.com). The experiments were conducted in six biological replicates ($n = 6$).

### Biofilm formation assay

Overnight cultures of bacterial strains (20 ml each) were grown in LB (DH5α and EDL933), LB supplemented with 30 μg/ml chloramphenicol (DH5α and EDL933 with pOX38 plasmid) or LB supplemented with 100 μg/ml streptomycin (DH10β). On the next day, the cultures were centrifuged 20 min 2000 x g and resuspended in 50 ml fresh LB to remove antibiotics. Following an additional round of centrifugation, the resulting pellet was resuspended in 5 ml fresh LB without antibiotics. For measurement of biofilm formation in increasingly dynamic

environments, 100 μl of each donor strain (F⁺) or plasmid-cured strain (F⁻) was combined with 1 ml of DH10β recipient strain and mixed thoroughly by pipetting. Resulting inoculums were evenly distributed into 3 glass tubes with 3 ml LB for each of the tested condition. For steady condition, the culture was incubated in a stationary incubator, the shaken and glass beads conditions were incubated with 180 rpm. shaking, the latter was supplemented in 1.5 g of sterile glass beads (2-3 mm in diameter). The biofilm developed for 24 h at 37 °C with 180 rpm shaking. For quantification, resulting biofilms were carefully washed 5 times with 10 ml of water to remove unattached planktonic cells. Then, 1 ml of crystal violet was added and incubated on a roller for 10 min to allow the stain to cover biofilms. Excess crystal violet was removed, followed by additional 5 times 10 ml washes with water. Stained biofilms were solubilised by addition of 1 ml 30% acetic acid and vortexed until no crystal violet was left on the walls of the glass tube. The absorbance was measured on 10-fold dilutions of resulting crystal violet suspensions using a 96-well-plate reader at 595 nm wavelength. The experiments were conducted in three biological replicates followed by three absorbance measurements for each replicate ($n = 9$). Resulting values were plotted using GraphPad prism. Statistical significance was calculated by two-way ANOVA test with Tukey's adjustment for multiple comparisons. To visualise the bacteria within the biofilm matrix by negative-stain electron microscopy, 10 μl culture of bacterial cells prior to staining with crystal violet were applied onto glow-discharged 300 mesh grids and visualised on the electron microscope.

### Bacterial strain and growth conditions for force measurements

The *E. coli* JE2571 strain harbouring the F-plasmid, pED208 was resuscitated from frozen storage by streaking for well-isolated single colonies on Lysogeny broth (LB) agar, supplemented with 96 μ/mL 5-bromo-4-chloro-3-indolyl-β-D-galactopyranoside (X-gal). Following overnight incubation at 37 °C, a well isolated single dark blue pigmented colony was inoculated into 10 ml fresh LB and incubated at 37 °C for 18 h, with shaking at 160 rpm. The next day, bacteria was subcultured into 10 ml of fresh LB and further grown at 37 °C until an $OD_{600nm}$ of 0.6 to 0.8. Bacteria were maintained at 37 °C without shaking prior force extension measurements. Before the force extension measurements, 10 μl of the liquid culture was suspended in 1000 μl of ice-cold PBS. This was done to stop and slow down potential polymerisation or de-polymerisation during the measurements. We measure the F-pili force-extension response of 30 cells from 2 biological replicates.

### Negative stain TEM imaging of *E. coli* cells and purified F-pili

For negative staining of whole bacteria producing F-pili, 10 μl culture of bacterial cells (grown in LB until $OD_{600nm}$ of 1.5) was applied onto glow-discharged 300 mesh carbon-coated grids (Agar Scientific) and incubated for 2 min. Following two subsequent washes with 10 μl of water, the grids were stained with 0.2% (w/v) phosphotungstic acid for 7 s before blotting. The grids were then imaged on a Technai12 Spirit transmission electron microscope (FEI) operating at 120 kV and equipped with 2 K eagle camera (FEI). Purified F-pili were applied onto glow-discharged 400 mesh carbon-coated grids (AgarScientific) and incubated for 2 min prior two washes with 10 μl of water. The grids were stained with 2% uranyl acetate for 1 min, blotted with filter paper and imaged on a Technai12 Spirit transmission electron microscope (FEI) operating at 120 kV and equipped with 2 K eagle camera (FEI).

### F-pili purification from the bacterial cell surface

*E. coli* JE2571 harbouring the F-plasmid, pED208 were grown on large LB agar plates. The bacterial lawn was collected from the plate surface using SSC buffer (150 mM NaCl pH 7.2 and 15 mM sodium citrate). The cells were left resuspending for 2 h at 4 °C with gentle agitation, followed by two centrifugation at $10,800 \times g$ for 20 min. The supernatant

was precipitated using 500 mM NaCl and 5% PEG 6,000 and allowed to incubate for 2 h at 4 °C. The precipitate was separated from the mix by centrifugation at $15,000 \times g$ for 10 min after being resuspended in sterile water. The supernatant was precipitated for a second time as described above, and subsequently, centrifuged at $15,000 \times g$ for 20 min. The F-pili pellet was resuspended in PBS pH 7.5, and applied on a premade CsCl step gradient (1.0–1.3 g/cm³). After centrifugation at $192,000 \times g$ for 17 h at 4 °C, the fraction containing the F-pili was carefully removed and extensively dialysed against PBS pH 7.4.

### F-pili cryo-EM vitrification and imaging

The 4 μl of purified F-pili sample was deposited onto glow-discharged 400 mesh copper Lacey grids coated with a 2 nm layer of carbon. Grids were vitrified using a Vitrobot (FEI) operating at 100% humidity and 25 °C with a wait time of 1 min and blot time of 3.5 s before the grid is plummeted into liquid ethane for freezing. The images were collected on a FEI Technai G2 Polara operating at 300 kV with a Gatan K2 Summit direct electron detector equipped with a Quantum energy filter and an energy selecting slit width of 20 eV. The images were taken with a pixel size of 1.1 Å/pixel and with a total dose of 50 e⁻/Å² split over 60 frames. The movie frames were aligned and summed using the MotionCor2 software[46].

### F-pili thermochemical stability

Purified F-pili was incubated for 1 h in water, 1 M NaOH or 8 M Urea followed by boiling at 100 °C. Sample were imaged by TEM following the same procedure described above.

### Force measurements and sample preparation

Force spectroscopy measurements was performed using an in-house optical tweezers instrument build around an inverted microscope (Olympus IX71, Olympus, Japan) equipped with a high numerical aperture water immersion objective (model: UplanSApo60xWIR 60X N.A. = 1.2; Olympus, Japan)[47]. To trap cells and microspheres a 2.0 watt DPSS laser (Rumba, Cobolt AB, Solna, Sweden), which operates at 1064 nm in CW mode, was used and the position of the trapped object was monitored by projecting the beam of a low-power fibre-coupled HeNe laser (operating at 632.8 nm, 117 A, Spectra Physics/Newport, Irvine, CA) onto a position-sensitive detector (L20 SU9, Sitek Electro Optics, Partille, Sweden)[48]. This allows for sub-nm spatial resolution of a trapped object. Before making a flow chamber, a 1:500 suspension of 9.5 μm carboxylate-modified latex beads (product no.2-10000, Interfacial Dynamics, Portland, OR) we prepared in Milli-Q water. These larger-sized beads were used as mounts for bacteria to be far away from the coverslips to avoid any interactions with the surface. Ten μl of the bead-water suspension were dropped onto 24 × 60 mm coverslips (no.1, Paul Marienfeld GmbH & Co, Lauda-Köningshofen, Germany) and placed in the oven for 60 min at 60 °C to immobilise the beads to the surface. To facilitate adhesion of the bacteria to the beads, a solution of 20 μl of 0.01% poly-L-lysine (catalogue no. P4832, Sigma-Aldrich, St. Louis, MO) was added to the coverslips and thereafter placed in an oven for 45 min at 60 °C. Cover slides were thereafter rinsed gently to remove superfluous poly-L-lysine. A sample was prepared by adding a ring of vacuum grease (Dow Corning, Midland, MI) around the area containing the poly-L-lysine-coated beads. Five μl suspension of bacteria and a 5 μl suspension of probe beads (surfactant-free 2 μm white amidine polystyrene latex beads, product no. 3-2600, Invitrogen, Carlsbad, CA) were carefully dropped onto the area and then sealed by placing a 20 × 20 mm coverslip (no.1, Paul Marienfeld GmbH & Co, Lauda-Köningshofen, Germany) on top. A single bacterium was trapped and firmly mounted onto a poly-L-lysine-coated 9.5 μm bead, as described in[49]. Subsequently, a 2 μm amidine bead was trapped and the stiffness of the trap was calibrated using the power spectrum[50]. To find the stability of the setup and optimal calibration time an Allan variance assay was performed[51]. The

trapped bead was thereafter attached to a F-pili by moving the bead in proximity to the mounted bacterium. By optically zooming in to the live video, it was possible to observe when a bead interacts with a pilus. With the pilus attached to the bead we applied tensile force by translating the piezo-stage (PI-P5613CD, Physik Instrumente GmbH & Co, Karlsruhe, Germany) using an in-house designed LabView program.

### eWLC model fitting

To describe the force-extension relation of F-pili we used the elastic worm-like-chain model (eWLC), which describes a macromolecule as a continuous polymer in thermodynamic equilibrium[52]. The eWLC model is given by:

$$x = L_0 \left( 1 - \frac{1}{2} \left( \frac{k_B T}{FP} \right)^{\frac{1}{2}} + \frac{F}{K_0} \right). \tag{1}$$

where $x$ is the extension of the polymer, $L_0$ is the contour length, $k_B$ is Boltzmann's constant, $T$ the temperature in Kelvin, $F$ the force applied to the polymer, $P$ is the persistence length and $K_0$ is the stretch modulus. We fit this model to our experimental data, $n = 20$, in MATLAB using the FDFIT package[53]. Pili that we were able to get multiple force curves on were also included in the analysis. Thus, the total number of fits was ~50. However, measurements on ten bacterial cells resulted in unreliable force curves, these were not used in the analysis. To account for drifts in the force data we added a force offset as an additional fitting parameter. We estimated $L_0$ by retracting the pili until the measurement probe came into contact with the cell membrane and used this point as a reference. From the model parameters we could also calculate the bending stiffness $B_s$ using $B_s = k_B T P$ and spring constant $k$ of the pili as $k = K_0 L_0$. We report all model parameters as mean ± standard error.

### Molecular dynamics simulations

The initial structure for the F-pilus was taken from PDB ID 5LEG[10]. Simulations of the F-pilus with and without phospholipids present were carried out. In order to obtain an initial structure of reasonable size for simulation 11 layers of the basic pentameric protein building block of the filament was included, and 10 layers of the interacting lipid chains, leading to 55 protein subunits and 50 lipid molecules in the simulated systems. All molecular dynamics simulations and system preparation were carried out with the Amber20/AmberTools21 software[54]. Systems were parameterised using the program tLeap. Force field parameters for the protein were taken from FF14SB[55], water was parameterised using the TIP3P force field[56], and phospholipids were modelled with the Lipid21 force field[57]. Monatomic counterions were added to the systems in order to neutralise the overall system charge, and the ions used the Joung and Cheatham parameters[58]. The protein was solvated in a rectangular water box that was 12 Å in both the x and y directions, and 100 Å in the z direction. The filament is aligned along the z axis. The large water buffer in the z direction allowed for extension of the system during steered molecular dynamics simulations.

Systems were energy minimised over 5000 steps (3000 steps of steepest descent followed by 2000 steps of conjugate gradient). Restraints of 10 kcal mol⁻¹ Å⁻² were placed on the alpha carbon atoms during minimisation. Two stages of gradual heating were carried out. First, the system was heated from 0 K to 100 K over 20 ps, and then held at 100 K for 30 ps, while the volume was held constant. During this stage, 10 kcal mol⁻¹ Å⁻² restraints were placed on the alpha carbons and on lipid molecule heavy atoms (for systems without phospholipids, restraints were only placed on protein atoms). After this step, the system was further heated from 100 K to 310 K over 20 ps, and then for the next 80 ps the system was held at 310 K. This step was carried out at a constant pressure of 1 bar, and the same restraints were applied as in the first stage of heating. After heating, systems were equilibrated to

prepare them for production simulations. Equilibration was carried out over six stages. During the first five stages the alpha carbons as well all lipid molecule heavy atoms were restrained with a force constant that was gradually reduced (10.0, 5.0, 2.5, 0.5, and 0.1 kcal mol⁻¹ Å⁻² for each stage, respectively). Each of the first five stages of equilibration lasted for 200 ps. During the sixth stage of equilibration, the restraints were modified so that only the five protein monomers at the tip of the filament and the five at the base had their alpha carbon atoms restrained (except for the first 17 amino acids of each monomer), with the rest of the protein atoms allowed to move freely. Lipid molecules were allowed to move freely in the sixth equilibration phase. During the sixth stage, the restraint strength remained at 0.1 kcal mol⁻¹ Å⁻². The sixth equilibration stage lasted for 5 ns. Equilibration was carried out in the NPT ensemble at 1 bar and 310 K. The final coordinates from the end of equilibration were used to initiate the steered molecular dynamics simulations.

Steered molecular dynamics (sMD) simulations[59] were performed in the NPT ensemble using a temperature of 310 K and by setting the jar = 1 flag in Amber20[60]. In order to perform the pulling simulations, the z-distance between the alpha carbons of the bottommost protein pentamer and the topmost protein pentamer (except for the first 17 amino acids of each monomer) was chosen as the collective variable (CV). A restraint of 0.5 kcal mol⁻¹ Å⁻² was added to the alpha carbons of the bottommost protein pentamer (except for the first 17 amino acids of each monomer) to prevent overall translations and rotations of the system during sMD simulations. The fxyz = 1 option was used in Amber20 to apply the sMD force along the z-axis. Pulling speeds of 1 Å/ns were used for each system, as this speed allows for simulations to be accomplished in a reasonable amount of time, although the speed is still significantly faster compared to experimental extension rates. The resulting values of the applied force and the CV extension was saved every 2 ps. If the extended system came too close to its periodic image (i.e., coming closer to its image than approximately 12 Å), the sMD simulation was ended at that point.

In addition to sMD simulations, 100 ns long equilibrium simulations were carried out also starting from the last trajectory snapshot of the sixth equilibration stage. During these extended equilibrium simulations, the alpha carbons of the bottommost protein pentamer (except for the first 17 amino acids of each monomer) were restrained with a 0.1 kcal mol⁻¹ Å⁻² force constant. All other conditions remained the same as in the sixth stage of equilibration.

The integration timestep used for heating, equilibration, sMD and the extended equilibrium simulations was 2 fs. To achieve a 2 fs timestep, the SHAKE algorithm was applied to constrain hydrogen bonds[61]. The Langevin thermostat was used to regulate temperature during the simulations, with a 1 ps⁻¹ collision frequency[62]. The Monte Carlo barostat was used to maintain the system pressure at 1 bar in all NPT phases of the simulation[62]. Long range electrostatics were handled using the particle mesh Ewald method[63] and 8 Å was chosen for the real space interaction cut-off.

### Structure analysis and presentation

All analysis was performed using Visual Molecular Dynamics (VMD)[64] and AmberTools21[54]. Visualisation of simulations, included images and movie creation, were accomplished using VMD.

### The filamentous f1 phage infection assay

An overnight culture (500 µl) of *E. coli* harbouring the pOX38 plasmid (donor) was used to inoculate 20 ml of fresh LB supplemented with 30 µg/ml chloramphenicol. The culture was allowed to grow in 37 °C with shaking until $OD_{600}$ reached 0.6, then cells were collected by centrifugation 2,000 x g for 10 minutes and resuspended in 5 ml of fresh LB. 2 µl of a 10⁻⁸ dilution from a clarified f1 phage lysate was added and the solution was mixed thoroughly by pipetting. The mixture was transferred into two separate tubes, 2.5 ml each, and one was left in a

static incubator, while another was shaken at 180 rpm. Both infection reactions were allowed to proceed in 37 °C for 1 h. To assess the rates of infection, 400 μl of each condition were taken out at set time points: 5 min, 30 min, 1 h, then mixed with 3 ml LB-agarose (prepared as LB with 0.4% (w/v) agarose) and evenly spread on "bottom agar"−LB agar supplemented with 30 μg/ml chloramphenicol, following the standard double-layer plating procedure[65]. Plates were left to dry for 20 min and incubated overnight in 37 °C, forming well-defined plaques on the next day.

## Secondary structure content of simulated F-pilus models

Secondary structure content is calculated using the DSSP algorithm as implemented in AmberTools22. For the initial structure, the secondary structure percentage is calculated from the initial experimental structural snapshot. For the equilibrium simulations, the secondary structure percentages are calculated using the last 20 ns of data from each of the 100 ns equilibrium simulations. For the steered MD simulations, the data between 9 ns and 11 ns is used, and all five runs for each case (with lipids or without lipids). Structural samples were taken at 50 ps intervals in each case. The tip pentamer and base pentamer pilin subunits were not included in the calculations.

## Percentage of time for salt-bridge presence

Salt bridges were calculated using the VMD Salt Bridges Analysis Tool. A cutoff of 3.2 Å was used between the oxygen and nitrogen atoms of the participating amino acids, which is the default setting in the analysis tool. The calculations were performed for the equilibrium simulations with and without lipids, using the entire 100 ns data set, and using structural samples taken at 100 ps intervals. Salt bridges from or to the tip pentamer and base pentamer pilin subunits were not considered. The percentages are an average value over all of the remaining protein-protein salt bridge interactions in each case.

## Reporting summary

Further information on research design is available in the Nature Portfolio Reporting Summary linked to this article.

## Data availability

Source data are provided with this paper.

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

## Acknowledgements

J.L.B. acknowledges support under NSF Grant MCB–1817670. J.L.B. also acknowledges use of the Electronic Laboratory for Science and Analysis (ELSA) high performance computing cluster at The College of New Jersey for conducting the simulations reported in this paper. This cluster is funded, in part, by the NSF under grants OAC–1826915 and OAC–1828163. J.L.B. also thanks Dr. Callum Dickson for early access to the Lipid21 parameters through his GitHub page, and for helpful discussion about use of the parameters. We thank Paul Simpson (Electron Microscopy Facility, Centre for Structural Biology, Imperial College London) for in-house EM support. We thank Prof. Jose Penadés and Prof. Alain Filloux (MRC Center for Molecular Bacteriology and Infection) for general insights and critical reading of the manuscript. We also thank Prof. Sivaramesh Wigneshweraraj (MRC Center for Molecular Bacteriology and Infection), Prof. Marjanca Starčič Erjavec (University of Ljubljana) and Dr. Alvaro San Millán (Centro Nacional de Biotecnología, Madrid) for providing bacterial strains. This work was supported by the Swedish Research Council (2019-04016) to M.A. and Welcome Trust Award 215164/Z/18/Z to T.R.D.C.

## Author contributions

Conceptualisation, T.R.D.C., M.A; methodology, J.B.P., T.D., J.L.B., T.R.D.C.; investigation, J.B.P., H.A., D.K.G., T.D., J.L.B. and T.R.D.C.; resources, M.F., J.L.B., M.A. and T.R.D.C.; writing—original draft, J.B.P., M.A., J.L.B. and T.R.D.C.; writing—review and editing, J.B.P., T.D., D.K.G.,

S.V., J.L.B., M.A. and T.R.D.C.; funding acquisition, J.P.V., J.L.B., M.A. and T.R.D.C.; supervision, M.A. and T.R.D.C.

## Competing interests

The authors declare no competing interests.
