## [Peer Review File · Nature Communications]

The F-pilus biomechanical adaptability accelerates conjugative dissemination of antimicrobial resistance and biofilm formationReviewer #1 (Remarks to the Author):

The work presented by Costa and colleagues on the F pilus is important. Molecular dynamics simulations can contribute substantially to the understanding of the mechanics of such systems. A lot of simulations were done on this large system, and 55 subunits were simulated, and 5 replicates of the simulations were performed. Simulations with and without phospholipids were performed to understand the role of these molecules.

Unfortunately, the presentation of the molecular dynamics results is limited to a few snapshots of the trajectory and an RMSD plot. This does not allow us to understand the processes of stability or unfolding at any interesting detail. The authors could have analyzed persistence of secondary structure, networks of intra or inter subunit interactions (hydrogen bonds, salt bridges), or distances between subunits.

For example, the differences between equilibrium simulations with and without phospholipids are documented essentially by showing snapshots of the simulations. This does not exploit in any way the information content of the simulation. Which interactions are affected by the presence or absence of phospholipids?

As regards the methodology itself, there is no justification for the pulling rate chosen. Also, one could have used different rates for the different replicates, to observe eventual trends and still get some statistics. Unfortunately, however, the statistics over the 5 separate simulations is not exploited at all.

Reviewer #2 (Remarks to the Author):

In the manuscript "The F-pilus biomechanical adaptability accelerates conjugative dissemination of antimicrobial resistance and biofilm formation", Patkowski and co-workers investigate the structural adaptation of the F-pilus and related functions using a combination of experimental, biophysical and molecular dynamics methods.

Conjugation is a mechanism of horizontal gene transfer where the DNA is transferred from a donor to a recipient cell. In Gram- bacteria, plasmid conjugation involves a type 4 secretion system associated with a conjugative pilus exposed at the surface of the donor cell. The conjugative pilus contacts the recipient cell to establish the mating pair before transfer, leading to cell aggregation in liquid conditions. If we now have a well-documented knowledge of the pilus' structure and composition, the way it allows mating pair formation and the transfer of single-stranded DNA and protein (the relaxase is covalently bound to the transferred ssDNA) is still elusive. Therefore, this work focuses on an important biological question using an original approach. Gaining insights into these questions is key to better understanding the transmission of genetic elements by conjugation and the dissemination of bacterial properties such as drug resistance.

Building from our current and thorough knowledge of the pilus' structure, the authors address the pilus' physical properties and role in transfer in hydrodynamics conditions and biofilm formation. Most importantly, they address the role of the phospholipid (phosphatidylglycerol) molecule associated with the pilin and coating the inner lumen of the pilus. They propose that phosphatidylglycerol is a crucial determinant of the pilus robustness, structure adaptability and function eventually.

This reviewer would like to emphasize several issues in the presentation and interpretation of the data. These issues are mainly related to the microbiology experiments corresponding to Figure 1. Since the interpretation of the effect of conditions (with or without shaking and beads) on conjugation and biofilm is the basis of the article and is not convincingly supported (according to this reviewer's consideration), this reviewer cannot recommend this article for publication. Nonetheless, the biophysical and molecular dynamics approach appears to provide relevant insights into the pilus biophysical properties and should be valued as such. Any links with the pilus function in conjugation and biofilms cannot be put forward in the current state of this work.

Concerns:

First and crucially, the results interpreted as conjugation efficiency (figure 1B) are actually the number of transconjugants in cfu (as indicated in the corresponding figure legend). It is difficult from the figure or the material and methods to understand what volume of conjugation mix was plated to obtain this indicated number of transconjugants. At least, the data plotted should be transconjugants cfu per ml of conjugation mix. But even in that case, the conjugation efficiency could not be compared between conditions. Indeed, the four conditions used to evaluate conjugation efficiency (Figure 1A: steady, shaking, glass beads and vortexed) are expected to affect cell growth as well (as confirmed in Figure S1). For these reasons, conjugation efficiency is conventionally presented as the number of transconjugant cells over donor cells (T/D) or as the number of transconjugants over the recipient population (T/T+R). Especially since the authors appear to have the data. Only in that case could the authors show convincingly that the variations in transconjugant frequencies are due to variations in transfer rather than differential growth.

Second, the Figure 1E shows biofilm formation in shaking conditions. The mix DH5aF+/DH10BF- shows more biofilm formation than the same mix in the same conditions in Figure 1C. Why is that? Is there a piece of information missing? Also, Figure 1E shows that conjugation mix forms more biofilms than donors or recipients alone. This is expected since the newly formed transconjugants (representing part of the recipient population, itself representing 90% of the conjugating population, Ratio D1:10R) will rapidly contribute to the biofilm.

Third, the phage infection test shows that cells infection by f1 increases in shaking conditions. The authors conclude, "elevated flow forces clearly accelerate the phage infection rate". This reviewer disagrees that these results reflect any acceleration of the infection but rather phage attachment efficiency. This reviewer thinks phage attachment depends on the probability of encounter between phage and pili, which is expectedly increased under shaking conditions. This reviewer disagrees even more with the paragraph's overarching conclusion: "These results confirm that it is solely the operation of the F-pilus that accounts for the higher conjugation efficiency and biofilm formation ability observed in dynamic media."

Figure 1D (I175) is cited after Figure 1E (I152) in the main text.

Word missing in the material and methods.

P3. I49-53

The authors state: "...the transport of plasmid DNA relies on a type IV secretion system (T4SS) to elaborate long extracellular conjugative pili. These long, helical appendages are critical to establish and maintain direct contact between two mating cells before the transport of plasmid DNA is initiated". This is not true for all conjugation systems. Some Gram+ bacteria do not use pili for conjugation and many plasmids from Gram- bacteria encode short pili that cannot be described as "long extracellular". Consequently, this reviewer recommends indicating in the introduction that the authors describe pili such as encoded by F-like plasmids or make the introduction more general to account for the variety of pili involved in conjugation.

P3-4. I66-69

The authors write: "IncF plasmids are present in common enteropathogenic species such as E. coli, Salmonella enterica, K. pneumoniae and Shigella spp. and are able to mobilise within the mouse gut microbiome at rates as high as those obtained under in vitro conditions" citing reference 16. However, reference 16 shows that "...that transfer occurs at a much lower rate in intestinal extracts than in laboratory media".

The lesser efficiency of R1 of F conjugation in vivo than in vitro was also reported in articles such as Neil et al., [PMID: 32963323]. Reference 16 actually shows that in vivo transfer is similar to that observed in biofilms in vitro.

The authors should re-phrase this part of the introduction, which is misleading in the present form.

P5. L99-101:

The authors write, "Conjugation has been generally considered a fragile process that needs to be conducted in stationary conditions to avoid disrupting the formed mating bridges between

bacteria. This, however, is a poor representation of how conjugation actually occurs in the unstable conditions present in the human gut or in nature". This statement is an oversimplification of the "general consideration" as it is well known that the pilus mediates the formation of mating pairs and mating aggregates that withstand vigorous dilution. This has even been used to address aggregate stability in early pioneer works (PMID: 357413 and PMID: 2880557).

Reviewer #3 (Remarks to the Author):

This is an interesting paper that describes how structural adaptations to the F-pilus impact its function. The authors reveal that phosphatidylglycerol molecules in the F-pilus confer structural stability that contributes to DNA transfer and biofilm formation.

Overall, I am positive about the study. There are however several points that require attention to detail as outlined below.

Specific comments.

L69-72. The statement is incorrect. The high prevalence of IncF plasmids in bacterial isolates that cause infections in the bloodstream and urinary tract does not infer an adaptation to these environments. There is no evidence that the F-pilus contributes to pathogenesis in these niches. Rather, the F-pilus likely functions by transferring antibiotic resistance plasmids in bacterial reservoirs such as the gut, which seed infection of the bloodstream and urinary tract.

L106. The pOX38 plasmid should be described.

L187-189. The authors switch from using pOX38 to pED208 without providing any rationale. This is an unfortunate omission since the plasmids are very different. Plasmid pED208 contains a traY mutation and constitutively expresses the tra genes – thus it is derepressed for conjugation. This should be acknowledged and referenced appropriately. The comment that TEM imaging revealed numerous F-pili at the cell surface needs to be taken in context – it is expected, but currently this would not be obvious to a general reader.

Were cells containing pOX38 examined by TEM? Were F-pili observed? If not, this could be stated as a lead into the switch of plasmids.

L340-343. The observation that pOX38 conjugation occurs equally on agar and in dynamic broth conditions has been observed previously and should be referenced (PMID: 32963323).

L367-375. I think this section could be toned down. If the model is partly based on a plasmid system derepressed for conjugation, this could be an alternative reason to explain functional differences relating to conjugation and biofilm formation.

Reviewer #4 (Remarks to the Author):

Patkowski et al. report on the biomechanical properties of conjugative pili, and the influence of hydrodynamic conditions on F-pilus mediated processes. They find, unexpectedly, that high hydrodynamic stress and shear forces increase conjugative activity. They furthermore show that the presence of F pili and increased conjugation under high hydrodynamic stress enhance biofilm formation. Furthermore, using optical tweezers, the authors nicely document the mechanical properties of F pili. This is a novel and valuable contribution to understanding the biology and mechanistic properties of bacterial conjugation. Experiments and simulations are nicely performed and sound. I'm supportive of publication but feel the authors go too far in some of their conclusions, however. These would need a more careful formulation or require additional experiment.

Main points

1. Experiments shown in Figure 1C and 1E convincingly show that increased conjugative activity increases biofilm formation. In the introduction to this paragraph (lines 133 – 137) the author suggest this may be by virtue of the conjugative pili acting as adhesive appendages. This may be plausible but is not directly shown. It's equally likely that other contact- or quorum-dependent signals contribute to the increased biofilm formation and may even be dominant compared to a (temporary?) connecting activity of conjugative pili. Addition of a TEM/SEM image of the resulting biofilm could perhaps inform on the level of connectivity and the presence of extracellular matrix between cells in the biofilms. The pellicle formation on the vessel wall is suggestive of ECM formation. To avoid over interpretation, it would be good to add a brief cautionary note regarding the adhesive role of conjugative pili, and tune down comments like Ln 177-179 "These results confirm that it is solely the operation of the F-pilus that accounts for the higher conjugation efficiency and biofilm formation ability observed in dynamic media."

2. The authors show the elastic behavior of F pili and characterize their mechanical properties. In Ln 259 – 261 they discuss pilus stretching: "This elasticity is seen in Figure 3, where we can document that the force increases linearly after 0.7 μm extension up to approximately 50 pN, in which the extension is stopped." Quoting a fixed maximal extension, as currently described and shown in the Figure 3B is confusing. So is illustrating the maximal extension using an atomic model of a small fragment of the fiber. I would expect a maximal extension per subunit and thus a proportionality to pilus length (i.e. 16% mentioned in line 259)? Is the quoted maximal extension for a single measured pilus, or are all pili of equal length and/or caught by the probe beat at a fixed distance from the cell? Or if the authors measure an average maximal extension of 0.7 μm , can they exclude this extension reflects a deformation from the cell to which the pili are attached?

3. In their discussion, the authors propose a model where increased mixing and the presence of conjugative pili accelerate expansion of biofilms by recruitment of planktonic, with sterile and conjugative F pili on edge cells catching new cells. I don't think such model can be concluded from the present data. The authors clearly show, unexpectedly, that high hydrodynamic environments increase conjugative activity. They also show that the presence of conjugative plasmids increases biofilm formation. From the data it is clear that the presence of F pili increases at least the frequency of cell-cell contacts in higher hydrodynamic environments, and the authors also show the high tensile strength of the conjugative pili. There is however no data on the longevity of F-pilus mediated contacts, or data on the structure of the biofilms and the relative contribution of adhesive structures versus secreted extracellular matrix components, nor is there data to say whether biofilms expand by cell proliferation within, by recruitment of planktonic cells, or by aggregation of smaller cell aggregates. It is equally possible that the higher frequency, possibly longer lifetime of cell-cell contacts results in the induction of contact- or quorum-dependent biofilm formation pathways. Additional experiments would have to be included to hold up the proposed model or a more open discussion should be formulated.

Minor comments.

- Figure 1E is cited in text before Figure 1D.
- Ln 187-189 The first mention of pED208 should include a reference to the plasmid.
- Ln 230 "Furthermore, we could not observe any conformational changes in the pili for forces up to 100 pN." I suggest to add "irreversible" or "inelastic" conformational changes. I don't think it can be excluded that the elastic stretching does not involve reversible conformational changes in the pilus subunits or global pilus architecture.
- Figure 4. For ease of understanding, I would suggest to show the native PG bound condition before the PG-less condition, i.e. replace A and B, and C and D. Same for panel A and B in Figure 5. Also the Figure caption for Figure 4 could be more clear. The force versus time curve is not so trivial to understand. It should be mentioned this reflects force on tip subunit pentamer in a SMD simulation.

Han Remaut

Reviewer #1 (Remarks to the Author):

The work presented by Costa and colleagues on the F pilus is important. Molecular dynamics simulations can contribute substantially to the understanding of the mechanics of such systems. A lot of simulations were done on this large system, and 55 subunits were simulated, and 5 replicates of the simulations were performed. Simulations with and without phospholipids were performed to understand the role of these molecules.

Unfortunately, the presentation of the molecular dynamics results is limited to a few snapshots of the trajectory and an RMSD plot. This does not allow us to understand the processes of stability or unfolding at any interesting detail. The authors could have analyzed persistence of secondary structure, networks of intra or inter subunit interactions (hydrogen bonds, salt bridges), or distances between subunits.

For example, the differences between equilibrium simulations with and without phospholipids are documented essentially by showing snapshots of the simulations. This does not exploit in any way the information content of the simulation. Which interactions are affected by the presence or absence of phospholipids?

In order to probe the dynamics simulations further we have carried out secondary structure calculations, and have also analysed two salt bridges that have been previously identified from the cryo-EM structure as being present in the model, to understand the model's behaviour with and without the PG lipids present (New supplementary Table 1).

First, we have calculated the total secondary structure content of the filament over time, not including the top and bottom pentamer in the calculation, within the equilibrium simulations, and also for the initial experimental structure. The secondary structure content is calculated using the DSSP method as implemented in the AmberTools analysis software cpptraj. The calculations are performed over the last 20 ns of each of the equilibrium simulations with and without lipids present using frames from the trajectories sampled at 50 ps intervals (and for the initial structure, it is calculated just for that single snapshot). We observe that in the initial structure the content is primarily alpha helix. This predominance of alpha helix remains in both the simulations with and without lipids, and the total alpha helical content increases by between 3-7% compared to the initial structure. We note that the simulation without lipids increases in alpha helix content and total helical (defined as the sum of alpha, 3-10, and pi helix) content slightly less compared to the simulation with lipids. Therefore, our data suggests there is not a strong effect on the secondary structure from the lipids.

We have now also evaluated some subunit-subunit salt bridge interactions in our analysis that have been identified as important interactions between the proteins in the filament based on previous structural work (New Supplementary Table 2). In particular, we focused on a known salt bridge interaction present between adjacent subunits in the filament towards the interior filament surface (Glu29-Lys41 of nearby chains) and a salt bridge interaction present between adjacent subunits in the filament on the exterior filament surface (Asp18-Lys64). We calculate the presence of this salt bridge in our equilibrium simulations over the entire 100 ns trajectory (filtering the data to snapshots every 100 ps) and report the overall average percentage of its presence in the simulations with and without lipids present. We calculated these interactions except for those involving the tip pentamer and base pentamer subunits. For the Glu29-Lys41 interaction we observe that there is an overall reduction in the percentage time that the salt bridge is present in the simulation when lipids are removed from the model. For the Asp18-Lys64 interaction, which is on the exterior filament surface, we find that the overall presence of the salt bridge is low in both the simulation with and without lipids, with only a very modest reduction in the presence of the salt bridge without lipids. The exterior surface salt bridge (Asp18-Lys64) is significantly more exposed to the external solvent, and so we expect that the lower

overall percent presence of this salt bridge compared to the Glu29-Lys41 interaction in the filament interior is to be expected. The cutoff used for defining a salt bridge was 3.2 Å between the oxygen and nitrogen of the two interacting amino acids, which is the standard implementation in the VMD Salt Bridges analysis tool.

We have included information about the secondary structure and these salt bridging interactions in the text (Lane 312-331, 415-419), and as Supplementary Tables 1 and 2.

As regards the methodology itself, there is no justification for the pulling rate chosen. Also, one could have used different rates for the different replicates, to observe eventual trends and still get some statistics. Unfortunately, however, the statistics over the 5 separate simulations is not exploited at all.

We thank the reviewer for this comment. The pulling rate of 1 Å/ns used in the steered MD simulations is a typical “low velocity” pulling speed used in simulations of large filamentous structures (e.g., references 1 and 2 below), which provides a trade-off between the simulation time required for the calculation and modest simulated pulling speeds. Even at 1 Å/ns, steered MD pulling speeds are 100,000 - 1,000,000 times faster than typical experimental pulling speeds using optical tweezers or AFM pulling techniques. Therefore, steered MD is considerably off-equilibrium and reducing the speed (and thereby increasing the simulation time) by even a factor of 5 or 10 compared to 1 Å/ns does not reach considerably closer to equilibrium than it warrants additional computational resources. Future work to further reduce pulling speeds in computational experiments would involve working to develop coarse-grained models, albeit at the expense of reducing the resolution of the model. Faster pulling speeds would be farther from equilibrium still, which is why we chose to explore 1 Å/ns in this particular set of experiments. We have added comments about the motivation for the choice of the pulling speed to the manuscript in the Methods section.

With regards to statistics over the pulling simulation, in particular in the simulations with no PG lipid molecules present, the nature of each of the disruptions of the filament are different enough (due to the stochastic nature of the steered MD simulations) where formal statistical averages are not immediately useful. However, we do believe that our depiction of the multiple force versus time curves presented in the paper already gives one way to interpret the variability across the pulling simulations already. Indeed, we also note that *in vivo* we never anticipate that the F pilus system is under forces that cause extension of the system to the point of structural failure. Nonetheless, we do think that the reviewer’s comment about secondary structure analysis would be interesting to analyse in the pulling simulations, and therefore we analysed how the pulling forces may influence the protein secondary structure over time with and without lipids. It seemed reasonable to perform the calculation by analysing the pulling simulation data between 9 and 11 nanoseconds (taking structural snapshots every 50 ps for the calculations) in each simulation (and again excluding the tip and base pentamers of subunits). This region of each force graph occurs right in the region of the force plateau, and so we are comparing similar points in time after the steering force has been applied for equal amounts of time. The averages presented in the table below are taken over all five of the simulations for each of the two scenarios (with and without lipids). Our calculations show that around the plateau simulated force in both the simulations with and without lipids present, the secondary structure is largely indistinguishable between the two sets of simulations. It is also very similar to the measured secondary structure from the corresponding equilibrium simulations. Therefore, the individual monomers of the F pilus appear to be resilient against changes to their secondary structure due to the application of external force. We have added comments (Lane 312-331, 415-419, 662-664) and the data (Supplementary Table 1) to the manuscript as well.

1) Dahlberg T, Baker JL, Bullitt E, Andersson M. Unveiling molecular interactions that stabilize bacterial adhesion pili. *Biophys J*. 2022 Jun 7;121(11):2096-2106. doi: 10.1016/j.bpj.2022.04.036. Epub 2022 Apr 30. PMID: 35491503; PMCID: PMC9247471.

2) Baker JL, Biais N, Tama F. Steered molecular dynamics simulations of a type IV pilus probe initial stages of a force-induced conformational transition. *PLoS Comput Biol*. 2013 Apr;9(4):e1003032. doi: 10.1371/journal.pcbi.1003032. Epub 2013 Apr 11. PMID: 23592974; PMCID: PMC3623709.

Reviewer #2 (Remarks to the Author):

In the manuscript "The F-pilus biomechanical adaptability accelerates conjugative dissemination of antimicrobial resistance and biofilm formation", Patkowski and co-workers investigate the structural adaptation of the F-pilus and related functions using a combination of experimental, biophysical and molecular dynamics methods.

Conjugation is a mechanism of horizontal gene transfer where the DNA is transferred from a donor to a recipient cell. In Gram- bacteria, plasmid conjugation involves a type 4 secretion system associated with a conjugative pilus exposed at the surface of the donor cell. The conjugative pilus contacts the recipient cell to establish the mating pair before transfer, leading to cell aggregation in liquid conditions. If we now have a well-documented knowledge of the pilus' structure and composition, the way it allows mating pair formation and the transfer of single-stranded DNA and protein (the relaxase is covalently bound to the transferred ssDNA) is still elusive. Therefore, this work focuses on an important biological question using an original approach. Gaining insights into these questions is key to better understanding the transmission of genetic elements by conjugation and the dissemination of bacterial properties such as drug resistance.

Building from our current and thorough knowledge of the pilus' structure, the authors address the pilus' physical properties and role in transfer in hydrodynamics conditions and biofilm formation. Most importantly, they address the role of the phospholipid (phosphatidylglycerol) molecule associated with the pilin and coating the inner light of the pilus. They propose that phosphatidylglycerol is a crucial determinant of the pilus robustness, structure adaptability and function eventually.

This reviewer would like to emphasize several issues in the presentation and interpretation of the data. These issues are mainly related to the microbiology experiments corresponding to Figure 1. Since the interpretation of the effect of conditions (with or without shaking and beads) on conjugation and biofilm is the basis of the article and is not convincingly supported (according to this reviewer's consideration), this reviewer cannot recommend this article for publication. Nonetheless, the biophysical and molecular dynamics approach appears to provide relevant insights into the pilus biophysical properties and should be valued as such. Any links with the pilus function in conjugation and biofilms cannot be put forward in the current state of this work.

Concerns:

First and crucially, the results interpreted as conjugation efficiency (figure 1B) are actually the number of transconjugants in cfu (as indicated in the corresponding figure legend). It is difficult from the figure or the material and methods to understand what volume of conjugation mix was plated to obtain this indicated number of transconjugants. At least, the data plotted should be transconjugants cfu per ml of conjugation mix. But even in that case, the conjugation efficiency could not be compared between conditions. Indeed, the four conditions used to evaluate conjugation efficiency (Figure 1A: steady, shaking, glass beads and vortexed) are expected to affect cell growth as well (as confirmed in Figure S1). For these reasons, conjugation efficiency is

conventionally presented as the number of transconjugant cells over donor cells (T/D) or as the number of transconjugants over the recipient population (T/T+R). Especially since the authors appear to have the data. Only in that case could the authors show convincingly that the variations in transconjugant frequencies are due to variations in transfer rather than differential growth.

We agree with the reviewer comment and now present our data in the suggested Transconjugants over Recipient population format (T/T+R) to account for any potential effect of bacterial cell growth during the 30 minutes of incubation. The new Figure 1B where the formula T/T+R was used to measure conjugation efficiency confirms our previous interpretation of the data. Additionally, following the suggestion by the editor, we expanded the scope of our study and observed a similar trend in conjugation efficiency when the clinical isolate *Escherichia coli* O157:H7 EDL933 was used as donor bacteria.

Second, the Figure 1E shows biofilm formation in shaking conditions. The mix DH5aF+/DH10BF- shows more biofilm formation than the same mix in the same conditions in Figure 1C. Why is that? Is there a piece of information missing?

We agree with the reviewer comment that the mismatch between experiments could be misleading. Therefore, we decided to combine Figure 1 C and E into one single panel (new Figure 1C). The new Fig.1C also includes the requested EDL933 clinical isolate. The EDL933 isolate shows a similar biofilm formation ability pattern as the DH5a, thus supporting our previous interpretation of the data and conclusion.

Also, Figure 1E shows that conjugation mix forms more biofilms than donors or recipients alone. This is expected since the newly formed transconjugants (representing part of the recipient population, itself representing 90% of the conjugating population, Ratio D1:10R) will rapidly contribute to the biofilm.

We agree with the reviewer comment and decided to remove non-mixed cultures results and only compare mixed populations i.e. donor/recipient (DH5 α F⁺/ DH10 β and EDL933 F⁺/ DH10 β) populations vs plasmid-cured donor/recipient (DH5 α F⁻/ DH10 β and EDL933 F⁻/ DH10 β) populations.

Third, the phage infection test shows that cells infection by f1 increases in shaking conditions. The authors conclude, "elevated flow forces clearly accelerate the phage infection rate". This reviewer disagrees that these results reflect any acceleration of the infection but rather phage attachment efficiency.

We agree with the reviewer comment that "phage infection rate" description can be misleading and "phage attachment efficiency" will better describe our observations. We updated the text and changed "phage infection rate" to "phage attachment efficiency".

This reviewer thinks phage attachment depends on the probability of encounter between phage and pili, which is expectedly increased under shaking conditions. This reviewer disagrees even more with the paragraph's overarching conclusion: "These results confirm that it is solely the operation of the F-pilus that accounts for the higher conjugation efficiency and biofilm formation ability observed in dynamic media."

We agree with the reviewer comment that the phage attachment depends on the encounter between phage and pili. The success of conjugation depends on a similar mechanism – but in this case on the probability of the encounter between a recipient cell and pili. As we observed an increased rate of conjugation in mixing condition, we used this assay as a proof-of-concept to directly show that mixing

indeed improves the probability of those encounters. We also agree that this result was indeed expected, and accordingly, moved the panel to Supplementary Information (new Supplementary Figure1 C).

Figure 1D (I175) is cited after Figure 1E (I152) in the main text.

Resolved.

Word missing in the material and methods.

Resolved.

P3. I49-53 The authors state: "...the transport of plasmid DNA relies on a type IV secretion system (T4SS) to elaborate long extracellular conjugative pili. These long, helical appendages are critical to establish and maintain direct contact between two mating cells before the transport of plasmid DNA is initiated". This is not true for all conjugation systems. Some Gram+ bacteria do not use pili for conjugation and many plasmids from Gram- bacteria encode short pili that cannot be described as "long extracellular". Consequently, this reviewer recommends indicating in the introduction that the authors describe pili such as encoded by F-like plasmids or make the introduction more general to account for the variety of pili involved in conjugation.

This has been changed and the new sentence reads "At the molecular level, the transport of plasmid DNA in Gram-negative bacteria relies on a type IV secretion system (T4SS) to elaborate extracellular conjugative pili. These helical appendages are critical to establish and maintain direct contact between two mating cells before the transport of plasmid DNA is initiated⁴⁻⁶". (Lanes 49-53)

P3-4. I66-69 The authors write: "IncF plasmids are present in common enteropathogenic species such as *E. coli*, *Salmonella enterica*, *K. pneumoniae* and *Shigella spp.* and are able to mobilise within the mouse gut microbiome at rates as high as those obtained under in vitro conditions" citing reference 16. However, reference 16 shows that "...that transfer occurs at a much lower rate in intestinal extracts than in laboratory media". The lesser efficiency of R1 of F conjugation in vivo than in vitro was also reported in articles such as Neil et al., [PMID: 32963323]. Reference 16 actually shows that in vivo transfer is similar to that observed in biofilms in vitro. The authors should re-phrase this part of the introduction, which is misleading in the present form.

This has been changed and the new sentence reads "IncF plasmids are present in common enteropathogenic species such as *E. coli*, *Salmonella enterica*, *K. pneumoniae* and *Shigella spp.* are able to mobilise within the mouse gut microbiome at rates similar to those observed in biofilms formed under *in vitro* conditions¹⁶". (Lanes 66-69)

P5. L99-101: The authors write, "Conjugation has been generally considered a fragile process that needs to be conducted in stationary conditions to avoid disrupting the formed mating bridges between bacteria. This, however, is a poor representation of how conjugation actually occurs in the unstable conditions present in the human gut or in nature". This statement is an oversimplification of the "general consideration" as it is well known that the pilus mediates the formation of mating pairs and mating aggregates that withstand vigorous dilution. This has even been used to address aggregate stability in early pioneer works (PMID: 357413 and PMID: 2880557).

This has been changed and the new sentence reads: "It is unclear how the mechanical stress experienced by the bacteria in the environment or human gut affects the formation of mating bridges and thus bacterial conjugation efficiency." (Lanes 100-102)

Reviewer #3 (Remarks to the Author):

This is an interesting paper that describes how structural adaptations to the F-pilus impact its function. The authors reveal that phosphatidylglycerol molecules in the F-pilus confer structural stability that contributes to DNA transfer and biofilm formation.

Overall, I am positive about the study. There are however several points that require attention to detail as outlined below.

Specific comments.

L69-72. The statement is incorrect. The high prevalence of IncF plasmids in bacterial isolates that cause infections in the bloodstream and urinary tract does not infer an adaptation to these environments. There is no evidence that the F-pilus contributes to pathogenesis in these niches. Rather, the F-pilus likely functions by transferring antibiotic resistance plasmids in bacterial reservoirs such as the gut, which seed infection of the bloodstream and urinary tract.

Thank you for pointing this out. This has been changed in the text and now reads: "Additionally, the prevalence of IncF plasmids in bacterial isolates from urine¹⁷⁻²⁰ and blood samples²¹⁻²⁴ may suggest that IncF plasmids could benefit from adaptations that allow for dissemination despite high hydrodynamic forces present in those liquid environments." (Lanes 70-73)

L106. The pOX38 plasmid should be described.

Thank you for pointing this out. This has been changed in the text and now reads: "Donor strains (designated as F⁺) harbour the conjugative pOX38 plasmid, which belongs to IncF family, and is an F-plasmid derivative with a chloramphenicol resistance cassette²⁷." (Lanes 107-109)

L187-189. The authors switch from using pOX38 to pED208 without providing any rationale. This is an unfortunate omission since the plasmids are very different. Plasmid pED208 contains a traY mutation and constitutively expresses the tra genes – thus it is derepressed for conjugation. This should be acknowledged and referenced appropriately. The comment that TEM imaging revealed numerous F-pili at the cell surface needs to be taken in context – it is expected, but currently this would not be obvious to a general reader.

Were cells containing pOX38 examined by TEM? Were F-pili observed? If not, this could be stated as a lead into the switch of plasmids.

We agree with the reviewer comment that the switch between plasmid requires additional explanation. Our rationale behind using one and the other were indeed dictated by the repression and derepression of the system. We chose the derepressed pED208 for our force measurements because the higher number of pilus at the bacterial surface facilitated the readout of the experiment. As for the molecular dynamics simulations, we utilised the pED208 pilus structure because of the higher resolution model obtained in comparison to the pOX38 pilus model. Importantly, both pilin molecules and appendages are structurally very similar (see Costa et al Cell 2016 and Supplementary Figure 3) and we do not anticipate any difference in their molecular behaviour. The repressed pOX38 plasmid was used for our conjugation and biofilm formation assays because it would better resemble the canonical F-plasmid.

Accordingly, we have added the following to the text: "The plasmid is a member of the IncF family isolated from *Salmonella typhimurium*²⁸ that expresses tra genes constitutively. For that reason, it

was more suitable for the F-pili purification and force measurements but not for our conjugation and biofilm formation analysis.” (Lanes 184-187)

L340-343. The observation that pOX38 conjugation occurs equally on agar and in dynamic broth conditions has been observed previously and should be referenced (PMID: 32963323).

Thank you for pointing this out. This has been changed in the text and now reads: “We observed that bacterial connections established by the F-pilus are significantly more resilient than anticipated resilient and do not require stationary conditions to be preserved, which is in line with the observation that pOX38-mediated conjugation occurs at similar rates in broth and on agar surface.³⁹” (Lanes 366-368)

Reviewer #4 (Remarks to the Author):

Patkowski et al. report on the biomechanical properties of conjugative pili, and the influence of hydrodynamic conditions on F-pilus mediated processes. They find, unexpectedly, that high hydrodynamic stress and shear forces increase conjugative activity. They furthermore show that the presence of F pili and increased conjugation under high hydrodynamic stress enhance biofilm formation. Furthermore, using optical tweezers, the authors nicely document the mechanical properties of F pili. This is a novel and valuable contribution to understanding the biology and mechanistic properties of bacterial conjugation. Experiments and simulations are nicely performed and sound. I’m supportive of publication but feel the authors go too far in some of their conclusions, however. These would need a more careful formulation or require additional experiment.

Main points

1. Experiments shown in Figure 1C and 1E convincingly show that increased conjugative activity increases biofilm formation. In the introduction to this paragraph (lines 133 – 137) the author suggest this may be by virtue of the conjugative pili acting as adhesive appendages. This may be plausible but is not directly shown. It’s equally likely that other contact- or quorum-dependent signals contribute to the increased biofilm formation and may even be dominant compared to a (temporary?) connecting activity of conjugative pili.

We thank the reviewer for this comment. The currently accepted model of the involvement of conjugative pili was proposed by Ghigo in 2001 (<https://www.nature.com/articles/35086581>), whereby conjugative pili facilitate adhesion of planktonic cells to cells present on stationary surfaces. In 2003 Reisner et al., observed that F-pilin deletion mutants ($\Delta traA$) developed thin unstructured biofilms, very much like those of plasmid-free strains and simultaneously showing that flagella, fimbriae, curli or adhesins were dispensable to maintain the biofilm-promoting effect of the F-pilus. (<https://doi.org/10.1046/j.1365-2958.2003.03490.x>) In 2008 and 2010 May et al., visualised biofilms formed by F plasmid-harboring *E.coli* that showed connections by the F-pili (<https://doi.org/10.1128/JB.00823-08>, <https://doi.org/10.1007/s00438-010-0571-2>). Moreover, in the 2010 paper May et al. show that presence of the pilus-tip specific bacteriophage f1 abolishes the effect of the F-pilus on biofilm formation. All this combined with the fact that no quorum-dependent mechanism has been observed to affect the F system, convincingly points that the observed effect is due to the F-pilus. Following the reviewer’s suggestion, we have added a TEM image that convincingly shows that F-pilus mediates bacterial contacts between bacteria in the biofilm matrix (Figure 1C).

Addition of a TEM/SEM image of the resulting biofilm could perhaps inform on the level of connectivity and the presence of extracellular matrix between cells in the biofilms. The pellicle formation on the vessel wall is suggestive of ECM formation. To avoid over interpretation, it would be good to add a brief cautionary note regarding the adhesive role of conjugative pili, and tune

down comments like In 177-179 “These results confirm that it is solely the operation of the F-pilus that accounts for the higher conjugation efficiency and biofilm formation ability observed in dynamic media.”

We agree with the reviewer comment that the addition of TEM images will be beneficial for the clarity of our manuscript, these have now been added to Figure 1 C. Following the reviewers suggestion we have tone down the comment to: “These results, together with our TEM images of bacteria within the biofilm matrix, suggest that is the operation of the F-pilus that accounts for the higher conjugation efficiency and biofilm formation ability observed in dynamic media.” (Lanes 171-174)

2. The authors show the elastic behavior of F pili and characterize their mechanical properties. In Ln 259 – 261 they discuss pilus stretching: “This elasticity is seen in Figure 3, where we can document that the force increases linearly after 0.7 μm extension up to approximately 50 pN, in which the extension is stopped.” Quoting a fixed maximal extension, as currently described and shown in the Figure 3B is confusing. So is illustrating the maximal extension using an atomic model of a small fragment of the fiber. I would expect a maximal extension per subunit and thus a proportionality to pilus length (i.e. 16% mentioned in line 259)? Is the quoted maximal extension for a single measured pilus, or are all pili of equal length and/or caught by the probe beat at a fixed distance from the cell? Or if the authors measure an average maximal extension of 0.7 μm , can they exclude this extension reflects a deformation from the cell to which the pili are attached?

We understand the confusion here since the previous text and figure 3, which showed maximal extensibility, were not clearly addressing our data in the first version of the manuscript. The aim of Panel B was to illustrate extension under tensile force, not to compare it quantitatively to the force-extension data in panel C, or to connect it to the F pilus maximal extensibility. We have now removed the wording, maximal extensibility, to avoid confusion.

However, regarding maximum extensibility, and as the review is pointing out correctly, it depends on the number of subunits and, thus, the total pilus length. We have now added that in the manuscript. We hope that the new text is more clear and not addressing that we are measuring the maximal extensibility of the pilus. (Lanes 231-234)

In addition, reaching the maximal extensibility of a pilus, the elongation just before it breaks, is indeed not possible in our experiments since this will most likely require close to, or maybe over, nN forces. Forces that are much higher than we can apply. Such high forces are also significantly higher than what a bacterial nano-motor responsible for polymerizing pili can achieve and are not relevant to our context (Merz, A. J., So, M. & Sheetz, M. P. Pilus retraction powers bacterial twitching motility. Nature 407, 98-102, doi:10.1038/35024105 (2000)).

It is a valid question that force measurements can deform the cell wall. However, tensile force in the range applied to pili using optical tweezers is not enough to significantly deform a bacterial cell's outer membrane since Young's modulus of an *E. coli* cell is several MPa. Therefore, we are confident that the force-extension data represents the F pilus response. This is also in line with previous studies in which force measurements have been done on both purified pili and pili attached to *E. coli* cells.

3. In their discussion, the authors propose a model where increased mixing and the presence of conjugative pili accelerate expansion of biofilms by recruitment of planktonic, with sterile and conjugative F pili on edge cells catching new cells. I don't think such model can be concluded from the present data. The authors clearly show, unexpectedly, that high hydrodynamic environments increase conjugative activity. They also show that the presence of conjugative plasmids increases biofilm formation. From the data it is clear that the presence of F pili increases at least the frequency of cell-cell contacts in higher hydrodynamic environments, and the authors also show the high tensile strength of the conjugative pili. There is however no data on the longevity of F-pilus

mediated contacts, or data on the structure of the biofilms and the relative contribution of adhesive structures versus secreted extracellular matrix components, nor is there data to say whether biofilms expand by cell proliferation within, by recruitment of planktonic cells, or by aggregation of smaller cell aggregates. It is equally possible that the higher frequency, possibly longer lifetime of cell-cell contacts results in the induction of contact- or quorum-dependent biofilm formation pathways. Additional experiments would have to be included to hold up the proposed model or a more open discussion should be formulated.

We agree with the reviewer comment that we do not show the longevity of F-pilus mediated contacts and therefore decided to remove our Figure 6 model from the manuscript. We have also formulated a more open discussion in the text. (Lanes 385-394)

Minor comments.

- Figure 1E is cited in text before Figure 1D.

Resolved.

- Ln 187-189 The first mention of pED208 should include a reference to the plasmid.

Thank you for pointing this out. This has been changed in the text and now reads: "The plasmid is a member of the IncF family isolated from *Salmonella typhimurium*²⁸ that expresses *tra* genes constitutively. For that reason, it was more suitable for the F-pili purification and force measurements but not for our conjugation and biofilm formation analysis." (Lanes 184-187)

- Ln 230 "Furthermore, we could not observe any conformational changes in the pili for forces up to 100 pN." I suggest to add "irreversible" or "inelastic" conformational changes. I don't think it can be excluded that the elastic stretching does not involve reversible conformational changes in the pilus subunits or global pilus architecture.

Resolved. (Lane 241)

- Figure 4. For ease of understanding, I would suggest to show the native PG bound condition before the PG-less condition, i.e. replace A and B, and C and D. Same for panel A and B in Figure 5. Also the Figure caption for Figure 4 could be more clear. The force versus time curve is not so trivial to understand. It should be mentioned this reflects force on tip subunit pentamer in a SMD simulation.

Thank you for pointing this out. This has been changed and the new Figure 4 legends reads: "Figure 4. Force versus time curves are shown for (A) the F pilus simulations without phospholipids and (B) the F pilus simulations with phospholipids for each of the five runs (run 1 - red, run 2 - blue, run 3 - black, run 4 - orange, and run 5 - purple). Four representative snapshots from a steered MD simulation (run 1) are shown for the F pilus structure from the system (C) without phospholipids and (D) with phospholipids. The method to measure the force required to extend the F pilus by pulling on the pentamer at the pilus tip is the steered MD approach, which is applied to the system as described in the Methods section."

Han Remaut

Reviewer #1 (Remarks to the Author):

The authors have responded to the questions and have considerably improved the analysis of the molecular dynamics trajectories.

Reviewer #2 (Remarks to the Author):

The authors have performed text modifications and present additional results that answer part of my initial concerns. Several statements have been moderated or made more rigorous and are now in better agreement with the literature or the results presented. The authors now use the formula $T/T+R$ to measure conjugation efficiency. The results presented in Fig. 1B appear to confirm previous data and interpretation. They also reproduce previous results using *Escherichia coli* O157:H7 EDL933 strain as donor bacteria, which is a clear improvement.

Nonetheless, this new version of the manuscript still raises several concerns (new or related to previous ones).

In the first version of the manuscript, two different results were shown for biofilm formation of the same mix in the same conditions (DH5aF+/DH10BF- in previous Figure 1C and 1E). The authors say that there was a "mismatch" between experiments and have decided to combine Figure 1 C and E into one single panel (new Figure 1C). Could they indicate if they have pooled/gathered the data points from the previous 1C and 1E in the new 1C?

Most importantly and quite unexplainably, there is a one-log difference between the biofilm formation values (A595) presented in the new 1C compared to the previous 1C and 1E. Data now range between ~ 0.05 and ~ 0.2 compared to 0.1 and 2 previously. What is the reason for these completely different values? Can the authors explain this rather dramatic change?

L156-157. "the F-pili are responsible for connecting bacteria in the biofilm matrix". The statement "responsible for" should be removed. TEM image (right panel) shows pili connecting cells indeed, but this does not demonstrate that F-pili are responsible for connection within biofilms (as in sufficient). In addition, the authors should present a TEM image with comparable cell densities for the inset left panel (1C).

This reviewer, however, agrees with the statement (L.173-175): "These results, together with our TEM images of bacteria within the biofilm matrix, suggest that it is the operation of the F-pilus that accounts for the higher conjugation efficiency and biofilm formation ability observed in dynamic media."

The definition of pOX38 (L108-109) "pOX38 plasmid, which belongs to IncF family, and is a derivative of the canonical F-plasmid with a chloramphenicol resistance cassette" is not very informative. It should be said somewhere that the pOX38 is a reduced form of the F plasmid (deletion of the large region using restriction enzyme, hindIII maybe) that retains autonomous conjugation ability.

Reviewer #4 (Remarks to the Author):

The authors satisfactorily address my concerns raised with the primary manuscript and I would support publication of the work.

Lines 256-261 describing the optical tweezer experiments hold one additional point that needs clarification in their final manuscript:

The authors write "Thus, the fit provides the parameter values for the persistence length and

stretch modulus, which, for these data, are 112 nm and 496 pN, respectively. The average values for all measured pili are 112 ± 7 nm and 279 ± 4 pN ($n = 30$), and we give a full compilation of the estimated measured parameters in Table 1. The variation in these estimated model parameters could be explained by the possibility of two or more pili attaching to the probe bead or that pili tangle with each other."

How can the average stretch modulus over 30 measured pili be 279 pN with a standard deviation of just 4 if one of the measured pili has a stretch modulus of 496 pN? The standard deviation would be 40 minimally. Even then, 496 pN would be an outlier. Why chose an outlier to show a representative force-extension / retraction curve in Figure 3?

Reviewer #1 (Remarks to the Author):

The authors have responded to the questions and have considerably improved the analysis of the molecular dynamics trajectories.

We thank the reviewer for the positive feedback and support for publication.

Reviewer #2 (Remarks to the Author):

The authors have performed text modifications and present additional results that answer part of my initial concerns. Several statements have been moderated or made more rigorous and are now in better agreement with the literature or the results presented. The authors now use the formula $T/T+R$ to measure conjugation efficiency. The results presented in Fig. 1B appear to confirm previous data and interpretation. They also reproduce previous results using *Escherichia coli* O157:H7 EDL933 strain as donor bacteria, which is a clear improvement.

Nonetheless, this new version of the manuscript still raises several concerns (new or related to previous ones).

In the first version of the manuscript, two different results were shown for biofilm formation of the same mix in the same conditions (DH5aF+/DH10BF- in previous Figure 1C and 1E). The authors say that there was a “mismatch” between experiments and have decided to combine Figure 1 C and E into one single panel (new Figure 1C). Could they indicate if they have pooled/gathered the data points from the previous 1C and 1E in the new 1C?

In the new version of the manuscript, we did not use the datapoints that were already presented in the old Figure 1C and 1E, thus the data was not pooled/gathered into the new figure. Rather, all the data points in the new version of the figure (new figure 1C) were obtained after a new set of experiments were performed. These new experiments were conducted after the Reviewer’s suggestion to calculate conjugation efficiency with the $T/T+R$ formula.

Most importantly and quite unexplainably, there is a one-log difference between the biofilm formation values (A₅₉₅) presented in the new 1C compared to the previous 1C and 1E. Data now range between ~0.05 and ~0.2 compared to 0.1 and 2 previously. What is the reason for these completely different values? Can the authors explain this rather dramatic change?

In biofilm formation assays, some of the obtained values were over the absorbance limit of the spectrophotometer (A₅₉₅ above 2.0) – making the measurements less reliable. To improve this, we decided to conduct measurements on 10x dilutions of crystal violet suspensions, which were further averaged by three separate measurements. We thank the reviewer for pointing this out and this has now been updated in the *Methods* section as follows: “The absorbance was measured on 10-fold dilutions of resulting crystal violet suspensions using a 96-well-plate reader at 595 nm wavelength.”. (Lanes 498-500)

L156-157. “the F-pili are responsible for connecting bacteria in the biofilm matrix”. The statement “responsible for” should be removed. TEM image (right panel) shows pili connecting cells indeed, but this does not demonstrate that F-pili are responsible for connection within biofilms (as in sufficient). In addition, the authors should present a TEM image with comparable cell densities for the inset left panel (1C).

This reviewer, however, agrees with the statement (L.173-175): “These results, together with our TEM images of bacteria within the biofilm matrix, suggest that it is the operation of the F-pilus that accounts for the higher conjugation efficiency and biofilm formation ability observed in dynamic media.”

We have now updated the statement, which now reads “Furthermore, whole-bacteria negative-stain transmission electron microscopy (TEM) images show an extensive network of F-pili connecting bacterial cells, which likely contributes to the formation of the biofilm matrix (Figure 1C, inset).” (Lanes 157-159)

The micrographs shown in Figure 1C were taken from samples of equal OD₆₀₀, ensuring comparable cell numbers. It is however difficult to obtain a micrograph that contains the same cell density between F- and F+ fractions, even despite the same number of bacterial cells. That is because while the F- cells remain evenly distributed across the grid, the F+ cells form bacterial clumps that cause an uneven distribution of bacterial cells. This results in empty patches of no cells and patches with many cells clumped together as opposed to the even distribution observed for the F- cells. Nevertheless, we have picked another combination of micrographs that shows a more uniform cell density between both pictures, as suggested (**New Figure 1C, inset**).

We would also like to thank the reviewer for agreeing with our main and leading conclusion that reads: "These results, together with our TEM images of bacteria within the biofilm matrix, suggest that it is the operation of the F-pilus that accounts for the higher conjugation efficiency and biofilm formation ability observed in dynamic media."

The definition of pOX38 (L108-109) "pOX38 plasmid, which belongs to IncF family, and is a derivative of the canonical F-plasmid with a chloramphenicol resistance cassette" is not very informative. It should be said somewhere that the pOX38 is a reduced form of the F plasmid (deletion of the large region using restriction enzyme, HindIII maybe) that retains autonomous conjugation ability.

This has been now changed in the text to: "Donor strains (designated as F+) harbour the conjugative pOX38 plasmid that belongs to the IncF family of conjugative plasmids and is a reduced form of the canonical F plasmid, generated by HindIII digestion and circularisation of the largest fragment that retains the conjugative ability, additionally a chloramphenicol resistance cassette was also incorporated." (**Lanes 107-111**)

Reviewer #4 (Remarks to the Author):

The authors satisfactorily address my concerns raised with the primary manuscript and I would support publication of the work.

Lines 256-261 describing the optical tweezer experiments hold one additional point that needs clarification in their final manuscript:

The authors write "Thus, the fit provides the parameter values for the persistence length and stretch modulus, which, for these data, are 112 nm and 496 pN, respectively. The average values for all measured pili are 112±7 nm and 279±4 pN (n = 30), and we give a full compilation of the estimated measured parameters in Table 1. The variation in these estimated model parameters could be explained by the possibility of two or more pili attaching to the probe bead or that pili tangle with each other."

How can the average stretch modulus over 30 measured pili be 279 pN with a standard deviation of just 4 if one of the measured pili has a stretch modulus of 496 pN? The standard deviation would be 40 minimally. Even then, 496 pN would be an outlier. Why chose an outlier to show a representative force-extension / retraction curve in Figure 3?

We understand why there was a confusion of the presented data. One part was that the reporting of values was not clear. We did actually mention, in both the method section and Table 1, that values are reported with standard error, not standard deviation. We have now clarified this in the main text also. (**Lane 629**)

More importantly, thanks to the very observant reviewer who noticed the anomaly in the reported values, we were puzzled and therefore investigated and reassessed the data. To our surprise, we found a script error in the Matlab code used for the analysis of the data in this project that did not compensate correctly for trap stiffness. We have rewritten the code and again checked the data using two fitting routines, one in-house algorithm, and a published one, to compare. They both agree. The new results are included in the updated manuscript. Note that the persistence length and bending stiffness are only marginally changed (112 to 104 nm, and 461 to 446 pNm²). However, the stretch modulus increased significantly (279 to 2330 pN), and so did the spring constant (0.2 to 1.7 pN/nm). This change is, in our opinion, reasonable. The new assessment shows that F pili is more

similar in terms of elasticity to other pili types. For example, Type IV has a spring constant of 2.0 pN/nm which is in the same order as F pili with 1.7 pN/nm.

In the updated manuscript, we have updated the numbers and data in the main text and table; updated figure 3 with a new fit; included a box plot figure in the supporting materials (Figure S5) to show statistics better, and rewritten the force spectroscopy section and method to be more clear. Also, we included references to force data of Type IV pili. (**Lanes 241-245, 259-271, 275-278, 617-621**)

Reviewer #4 (Remarks to the Author):

The authors carefully addressed my remaining concerns. The clarifications on the calculated mechanical properties and the addition of the scatter plots showing the data points of the individual measurements in Supplementary Fig. 5 resolve my previous concerns.